# Most discriminative stimuli for functional cell type clustering

**Max F. Burg**[1-3]**, Thomas Zenkel**[4,5]**, Michaela Vystrčilová**[2]**, Jonathan Oesterle**[4,5]**,
Larissa Höfling**[4,5]**, Konstantin F. Willeke**[1,2,6]**, Jan Lause**[3,7]**, Sarah Müller**[1,3,7]**, Paul G. Fahey**[8,9]**,
Zhiwei Ding**[8,9]**, Kelli Restivo**[8,9]**, Shashwat Sridhar**[10,11]**, Tim Gollisch**[10-12]**, Philipp Berens**[3,7]**,
Andreas S. Tolias**[8,9,13]**, Thomas Euler**[4,5]**, Matthias Bethge**[3,5]**, Alexander S. Ecker**[2,14]

[1] International Max Planck Research School for Intelligent Systems, Tübingen, Germany [2] Institute of Computer Science and Campus Institute Data Science, University of Göttingen, Germany, [3] Tübingen AI Center, University of Tübingen, Germany, [4] Institute of Ophthalmic Research, University of Tübingen, Germany, [5] Centre for Integrative Neuroscience, University of Tübingen, Germany, [6] Institute for Bioinformatics and Medical Informatics, Tübingen University, Germany, [7] Hertie Institute for AI in Brain Health, University of Tübingen, Germany, [8] Department of Neuroscience, Baylor College of Medicine, Houston, TX, USA, [9] Center for Neuroscience and Artificial Intelligence, Baylor College of Medicine, Houston, TX, USA, [10] University Medical Center Göttingen, Department of Ophthalmology, Germany [11] Bernstein Center for Computational Neuroscience Göttingen, Germany [12] Cluster of Excellence "Multiscale Bioimaging: from Molecular Machines to Networks of Excitable Cells" (MBExC), University of Göttingen, Germany [13] Department of Electrical and Computer Engineering, Rice University, Houston, TX, USA, [14] Max Planck Institute for Dynamics and Self-Organization, Göttingen, Germany. Contact: max.burg@bethgelab.org, ecker@cs.uni-goettingen.de

## Abstract

Identifying cell types and understanding their functional properties is crucial for unraveling the mechanisms underlying perception and cognition. In the retina, functional types can be identified by carefully selected stimuli, but this requires expert domain knowledge and biases the procedure towards previously known cell types. In the visual cortex, it is still unknown what functional types exist and how to identify them. Thus, for unbiased identification of the functional cell types in retina and visual cortex, new approaches are needed. Here we propose an optimization-based clustering approach using deep predictive models to obtain functional clusters of neurons using Most Discriminative Stimuli (MDS). Our approach alternates between stimulus optimization with cluster reassignment akin to an expectation-maximization algorithm. The algorithm recovers functional clusters in mouse retina, marmoset retina and macaque visual area V4. This demonstrates that our approach can successfully find discriminative stimuli across species, stages of the visual system and recording techniques. The resulting most discriminative stimuli can be used to assign functional cell types fast and on the fly, without the need to train complex predictive models or show a large natural scene dataset, paving the way for experiments that were previously limited by experimental time. Crucially, MDS are interpretable: they visualize the distinctive stimulus patterns that most unambiguously identify a specific type of neuron.

## 1 Introduction

Animals perceive the world through an intricate network of neurons in the visual system. These neurons are organized into cell types that can be identified by their distinct responses to visual stimuli. Such functional cell typing has been demonstrated in the mouse retina (Baden et al., 2016), where functional cell types also match in their gene expression and morphology (Goetz et al., 2022). In higher visual areas, it has proven difficult to identify cell types based on their response patterns, although recent attempts point at a mix of continuous and discrete functional types in mouse primary visual cortex (V1) (Ustyuzhaninov et al., 2020; 2022) and primate cortical area V4 (Willeke et al., 2023).

Functional cell type identification currently often requires domain knowledge to handcraft suitable visual finger-printing stimuli (Farrow & Masland, 2011; Baden et al., 2016; Franke et al., 2017)

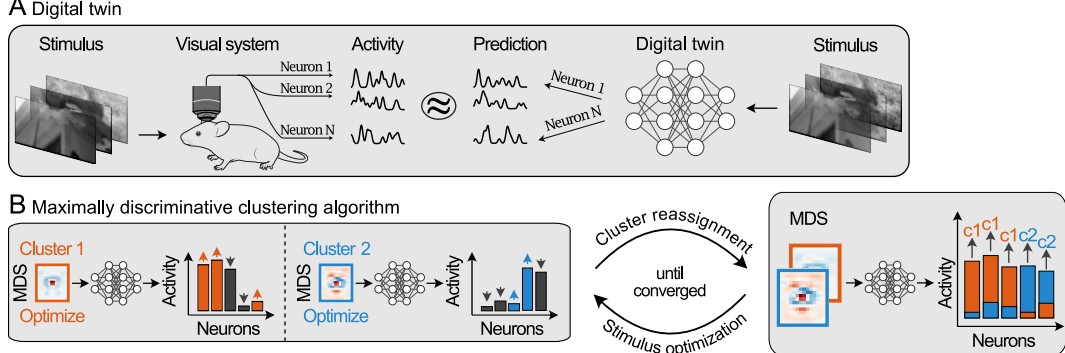

Figure 1: *Most discriminative stimulus (MDS) clustering based on a digital twin.* **A.** Digital twin model trained to mirror responses of neurons in the visual system. **B.** MDS clustering (Algorithm 1) iterates between optimizing MDS to drive neurons within one cluster while suppressing all others and reassigning neurons to the cluster associated with the MDS they respond most to.

and to select distinctive response features for classification (Farrow & Masland, 2011). This biases classification towards known cell types and might make it hard to discover new types. Also, in many brain areas the functional types are still unknown – hence no "ground truth" exists to tailor stimuli towards. Therefore, the study of both well known and less understood visual areas would benefit from a more neutral, data-driven approach that does not require domain knowledge. One idea would be to directly cluster cell's responses to unbiased naturalistic stimuli. However, the receptive field locations vary within a simultaneously recorded population of neurons, which means that each cell would "see" a different part of the stimulus. Hence, even cells with identical input-output function will respond differently to the same stimulus, preventing direct clustering of their responses.

Fortunately, resolving this issue has recently come within reach: artificial neural networks have been proposed as "digital twins" of the visual system – these networks receive visual stimuli as inputs and predict how a specific biological neuron would respond to that stimulus with high fidelity (Fig. 1A; Antolík et al. (2016); Batty et al. (2016); McIntosh et al. (2016); Klindt et al. (2017); Sinz et al. (2018); Walker et al. (2019); Lurz et al. (2020); Burg et al. (2021); Willeke et al. (2022); Ustyuzhaninov et al. (2022)). Such digital twins make it possible to analyze the computational properties of visual neurons *in silico*. For instance, digital twins allow to center a stimulus on each individual neuron's receptive field (Ustyuzhaninov et al., 2022) and to optimize it to maximally drive a single neuron, yielding its Maximally Exciting Input (MEI), that visualizes the neuron's preferred feature (Walker et al., 2019; Bashivan et al., 2019). Bashivan et al. (2019) attempted to make MEIs more specific by optimizing images that do not only drive *one neuron*, but also suppress *all* others. However, this is often not possible in practice, because multiple neurons of the same functional cell type are present, which respond similarly.

If we knew which cells belong to the same functional type, we could easily extend this approach and find *Most Discriminative Stimuli (MDS)* at the cell type level – stimuli that drive one *cell type* while suppressing all others when presented centered on each cell's receptive field. With the resulting MDS, we can discover the unique visual features that the different cell types preferably process, rendering MDS to be efficient finger-printing stimuli for easy cell type identification in future experiments. As it is unknown which neurons belong to which cell type *a priori*, we need to cluster cells into types and find the MDS for each type in a joint procedure.

Here, we tackle this task in an Expectation-Maximization (EM) style manner: We alternate between assigning neurons to cell type clusters, and optimizing a MDS for each cluster (Fig. 1B). We empirically validate our novel clustering algorithm by benchmarking the resulting clusters against well-established retinal ganglion cell (RGC) types (Baden et al., 2016; Field & Chichilnisky, 2007). We further demonstrate that our method found meaningful functional clusters when little domain knowledge is available, as in cortical area V4. Our clustering works across species (mouse, marmoset, macaque) and recording techniques (two-photon imaging, electrophysiology), and leveraging MDS as finger-printing stimuli can identify cell types 20% faster than traditional methods.

---

**Algorithm 1** Most discriminative clustering algorithm

---

Randomly assign neurons into clusters
**while** cluster assignments change **do**                                    ▷ Initial clustering
    M-step: Optimize MDS for all clusters with objective function Eq. (1)
    E-step: Re-assign each neuron to the cluster whose MDS drives it the most
**end while**
**while** cluster assignments change **do**                       ▷ Determine the optimal number of clusters
    **for** each cluster **do**
        Randomly assign this cluster's neurons into candidate sub-clusters
        Run EM clustering on sub-clusters until convergence
        Keep candidate sub-clusters if they improve mean objective $\langle J_c \rangle_c$ and remove empty clusters
    **end for**
**end while**
**return** Cluster assignments, most discriminative stimuli

---

## 2 RELATED WORK

**Cell type clustering.** Cell types form the building blocks of neural circuits across the brain (Sanes & Masland, 2015). To understand how these cell types perform computations in the visual system, a functional classification based on light responses may be most relevant (Vlasits et al., 2019). In the mouse retina, functional cell types have been studied extensively (Farrow & Masland, 2011; Baden et al., 2016; Franke et al., 2017), especially for retinal ganglion cells: RGCs can be hierarchically divided by response polarity into ON, OFF and ON-OFF types, and then further by response dynamics into transient and sustained types (Farrow & Masland, 2011). Using more complex stimuli, Baden et al. (2016) identified a finer cluster structure of at least 32 functional RGC types, many of which were later matched to morphological and genetic types (Bae et al., 2018; Tran et al., 2019; Goetz et al., 2022; Huang et al., 2022). As the Baden et al. (2016) functional clustering for RGCs is the most comprehensive available to date, we focus on their cluster labels in the retina experiments presented here. In cortex, functional cell types have been harder to define. For instance, the fingerprinting stimuli used in the retina rely mainly on temporal characteristics of full-field brightness changes or known properties such as direction selectivity probed with moving bars. How to find appropriate stimuli for the diverse spatial selectivity patterns observed in visual cortex (Walker et al., 2019; Ustyuzhaninov et al., 2022; Tong et al., 2023) is less clear. Ustyuzhaninov et al. (2020; 2022) modeled responses to natural images in mouse primary visual cortex (V1) with a convolutional neural network (CNN), where each neuron's response is modeled by a linear readout from a shared feature space. They then used the readout weights as a vector representation of the neuron's function. This approach captures the full selectivity of a neuron, but it requires showing natural scenes and training a predictive model to infer a neuron's cell type, whereas our approach can identify a neuron's cell type in new experiments quickly and without model training.

**Image optimization.** Optimizing input images has been used to maximize the response of units in artificial neural networks for object classification (Erhan et al., 2009; Zeiler & Fergus, 2014; Olah et al., 2017) and to distinguish between different classification models (Golan et al., 2020). Building on these ideas, Deep Neural Network (DNN) models of neural activity were used to generate maximally exciting inputs (MEIs) for biological neurons. These MEIs can drive single neurons *in vivo* (Bashivan et al., 2019; Walker et al., 2019; Franke et al., 2022; Hoefling et al., 2022; Willeke et al., 2023; Pierzchlewicz et al., 2023), even when using multiple diverse MEIs (Ding et al., 2023); inhibit a single neuron (Fu et al., 2023), or drive a single neuron but suppress others (Bashivan et al., 2019). While MEIs were recently extended to drive a predefined group of neurons (Ustyuzhaninov et al., 2022), there is no technique that at the same time suppresses the activity of all other neurons. Here, we provide this missing technique and extend it into a full-fledged clustering algorithm.

## 3 MOST DISCRIMINATIVE STIMULUS CLUSTERING ALGORITHM

Digital twins are models trained to predict neural activity based on shown stimuli. They allow us to search the stimulus space for most discriminative stimuli (MDS) that optimally separate functional groups of neurons in their activity (Fig. 1A). Here, we simultaneously generated MDS and clustered neurons in an EM-like fashion (Fig. 1B and Algorithm 1). We started by randomly assigning neurons

into a small number of initial "seed" clusters. Then we alternated between (M) optimizing an MDS for each cluster exclusively activating that cluster's neurons, while avoiding to activate other clusters, and (E) re-assigning neurons to the cluster whose MDS drove them the most (Fig. 1B). Periodically, we split candidate clusters into sub-clusters that we kept if this improved the objective (see below).

In each M step, we optimized an MDS $x_c$ (centered on the neuron's receptive field) to maximize the average response $\bar{r}$ of all neurons $j$ in the target cluster $c$, i.e. $\bar{r}_c(x_c) = \langle r_j(x_c) \rangle_j$, while suppressing the average response of all other clusters, $\bar{r}_k(x_c)$. We maximized the softmax-based objective

$$\max_{x_c} J_c = \max_{x_c} \left( \log \frac{\exp\left(\bar{r}_c(x_c)/\tau\right)}{\frac{1}{K}\sum_{k=1}^{K}\exp\left(\bar{r}_k(x_c)/\tau\right)} \right) , \tag{1}$$

where the temperature parameter $\tau$ determines how strongly we penalize large responses in other clusters (we set $\tau = 1.6$ which is the best within a broad optimum of the final objective, see Fig. 14). This yielded one MDS per cluster. In the E step, we re-assigned neurons to the cluster associated with the MDS to which they responded most strongly. We then alternated between the E- and M-step until cluster assignments did not change anymore.

To find the optimal number of clusters, we devised a procedure to determine if adding more clusters enhanced our overall clustering. We treated each cluster independently and split its neurons into several new sub-cluster candidates. Then we performed the EM clustering procedure on the sub-cluster candidates followed by a global optimization step on all clusters yielding optimal MDS and cluster assignments, and we kept new clusters if the mean objective across all clusters $\langle J_c \rangle_c$ improved (see Appendix A.1 for details). During the splitting procedure, we removed any clusters not containing any neurons. We then repeated cluster splitting until it did not improve the objective anymore. To finalize clustering, we optimized all MDS until convergence followed by a final reassignment step.

## 4 DATA AND IMPLEMENTATION DETAILS

**Mouse retinal ganglion cells.** We used a mouse retinal ganglion cell (RGC) dataset. A digital twin model for this dataset was previously published by Hoefling et al. (2022). The neurons in this dataset were stimulated with an ultraviolet (UV) and green channel naturalistic video (30 Hz frame rate). The digital twin was an ensemble of five CNNs that were trained to predict the time-varying RGC response with a 30 frame context window. Each member of the ensemble consisted of a core shared between all neurons and a neuron-specific linear readout (see Appendix A.2 for details). All models used in this study share this core-readout structure.

We assigned neurons to Baden et al. (2016) functional types using a classifier that predicts the cell type based on soma size and the light response to two synthetic fingerprinting stimuli, a "chirp" and a moving bar (Qiu et al., 2023). We removed neurons with less than $25\%$ assignment confidence and all functional types containing less than 50 modeled neurons, as they might have been underrepresented in digital twin training. Further, displaced amacrine cells, which are usually part of this dataset, were removed as well. This led to 2,448 RGCs of 17 types. To ensure MDS are not overfit to the cells they were optimized on and that they also work in future experiments, we computed MDS on a 80% training split of the cells and report results on the remaining held-out test cells. Splits retained relative functional group sizes. These type labels are no perfect ground truth, as they are subject to potential misassignments by the classifier.

**Stimulus optimization.** We synthesized MDS with the same parameters that Hoefling et al. (2022) used to optimize MEIs and summarize the key components here. We initialized the stimulus, a 50 frames $\times\,(18 \times 16)$ pixels $\times 2$ channels video-snippet, with Gaussian white noise. We fed this stimulus to the digital twin predicting a 20 frames response trace for each cell. The average predicted activity over the last ten frames was the input into the objective function (Eq. (1)). We optimized the stimulus by gradient ascent for ten steps with a learning rate of 100 (the optimal learning rate within the range $[10^{-3}, 10^3]$). During algorithm development we verified that sufficient optimization steps were performed by inspecting the objective values during optimization. To ensure that the MDS stayed within the pixel intensity range used when training the digital twin (Hoefling et al., 2022), we jointly renormalized both channels' pixel values of each frame to an $L_2$-norm of 30 after each optimization step. Additionally, we clipped the pixel values to the minimum (UV: $-0.913$, green: $-0.654$) and maximum (UV and green: $6.269$) light intensity of the stimulus projector that

displayed the videos, where zero corresponds to the mean luminance across the stimuli. This avoids that single pixels can exceed the display range of the projector and is a typical procedure in stimulus optimization settings (e.g. Walker et al. (2019); Hoefling et al. (2022); Willeke et al. (2023)).

**Simulation of experiment.** To estimate how time-efficient MDS are in determining a functional type, we did an *in silico* experiment with simulated observational noise. We modeled the trial-to-trial variability for each neuron by a gamma distribution whose mean we set to the digital twin's MDS response prediction for that neuron. We set the variance to be proportional to the mean and estimated the proportionality constant for each neuron from a subset of the stimuli in the dataset for which responses to three repeated stimulus presentations were provided. Then we sampled from the gamma distribution to generate 16 "trials" of activity and computed the neural responses by averaging the activity over the last ten frames (as in the stimulus optimization procedure). Model neurons were then assigned to the cluster whose MDS activated them the most.

**Marmoset retinal ganglion cells.** We additionally evaluated our approach on 167 marmoset RGCs obtained from Multi-Electrode Array (MEA) recordings of spiking activity in one retina while presenting a naturalistic gray-scale movie. For this data, no fine-grained reference taxonomy of functional types exists. We optimized the mouse digital twin (Hoefling et al., 2022) for this dataset (four CNN layers, larger 21 pixel spatial filters in the first layer; see Appendix A.3 for details).

**Macaque visual cortex area V4.** Additionally, we used a third dataset of extracellular multi-electrode recordings from macaque visual cortex area V4, covering spiking responses of 1,224 neurons to gray-scale natural images shown to awake macaque monkeys (Willeke et al., 2023). We kept only the 1,030 neurons for which Willeke et al. (2023) provided MEI-based cluster labels. Note that these single cell MEI clusters do not represent ground truth, as they were not verified with independent biological measurements. Again, we sampled train (80%) and test (20%) splits retaining relative cluster sizes. For digital twin details and MDS optimization parameters, see Appendix A.4.

## 5 RESULTS

**Mouse retinal ganglion cells.** We started by empirically validating our novel clustering algorithm on the mouse RGC dataset that was labeled with 17 well-established Baden et al. (2016) functional types. Our MDS clustering algorithm found seven functional clusters after discarding small clusters with less than ten cells. Clusters were reasonably well separated in terms of their mean response (Fig. 2A) to the MDS (Fig. 2B). While our MDS clustering yielded fewer clusters than that by Baden et al. (2016), the clusters we found comprised the well known functional cluster hierarchy of the retina (e.g. Baden et al. (2016); Farrow & Masland (2011)), namely OFF, ON-OFF, fast ON, and slow ON types, and one color-opponent type that was recently highlighted as having very distinct selectivity (Hoefling et al., 2022). Clusters differed in their preference for temporal frequencies, receptive field center sizes, strength of UV preference, and surround strength and structure (Fig. 2C).

Next, we compared the 17 Baden et al. (2016) types to our MDS clusters (Fig. 2D). We found that groups of Baden et al. (2016) types with similar functional properties matched well with our MDS clusters (color-coded labels in Fig. 2D based on clustering hierarchy in Baden et al. (2016)). Off-diagonal entries in Fig. 2A,D show that not all fine types could be perfectly separated by MDS. However, the block-diagonal structure (black boxes in Fig. 2D) indicates that cell types were only confused within groups of very similar function. All **OFF and ON-OFF types** (orange labels in Fig. 2D) mapped almost exclusively to MDS clusters 2 and 3, with a tendency of OFF cells mapping to cluster 2, and ON-OFF cells to cluster 3. From the groups of Baden et al. (2016) types, **fast ON types** ("ON loc tr OS", "ON trans", "ON high freq"; green in Fig. 2D) showed the clearest correspondence to a single MDS cluster, 38%, 93%, and 93% of cells of its three types mapping to MDS cluster 4, respectively. **Slow ON types** (blue in Fig. 2D) mapped mostly to clusters 5 and 6, with a preference for one of the two in most but not all types. In this group, one type ("ON DS sustained") even mapped exclusively to a single MDS cluster. Despite its name, the "OFF suppressed 1" type also mapped to ON MDS cluster 6, likely because this type has an ON response for some local stimuli (e.g. moving bar response in Baden et al. (2016)). The biggest exception of the slow ON types was "ON contrast suppressed", which did not map strongly to either cluster 5 or 6, but instead mapped to the color opponent MDS cluster 7 (Fig. 2D). Interestingly, this cluster showed the strongest one-to-one correspondence with a single Baden et al. (2016) type. The only other MDS cluster consisting of mostly one type was MDS cluster 1, which was dominated by

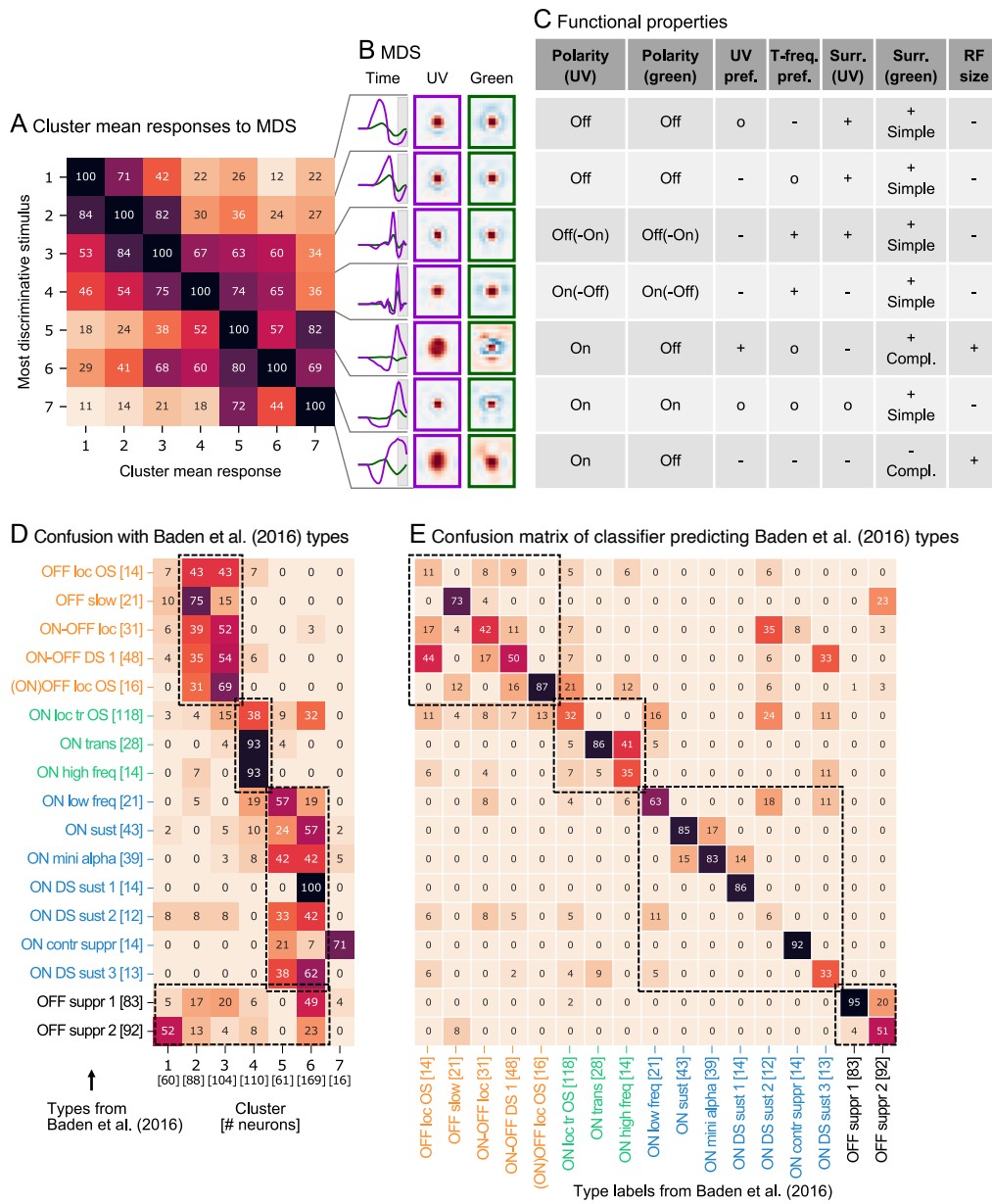

Figure 2: *Most discriminative stimuli clustering on mouse RGCs.* Results on held-out test neurons and empirical comparison to Baden et al. (2016) types shown. **A.** Cluster averaged digital twin response to the optimized MDS. Elements within a column normalized to highest value. **B.** MDS video snippets per cluster decomposed into time, UV- and green-channel stimulus components by singular value decomposition. For display, spatial components were re-scaled to a symmetric color scale between -1 (blue) and 1 (red). **C.** Summary of the functional characteristics for each cluster (UV-pref: UV-preference, T-freq. pref.: Temporal frequency preference; Surr.: Surround properties; RF: Receptive Field; $+/\circ/-$: High/medium/low). **D.** Confusion matrix between MDS clusters and Baden et al. (2016) types (predicted by a classifier for the mouse RGC dataset; Qiu et al. (2023)). Annotations in same color belong to the same hierarchical functional type. Elements within rows were normalized to sum to 1, annotated numbers display the number of neurons in percent. **E.** Confusion matrix of the classifier (Qiu et al., 2023) used to predict Baden et al. (2016) types for our dataset only having access to the same type of information as MDS clustering between Baden et al. (2016) type labels and predicted type labels on held out test data. Types confused by the classifier are merged into MDS clusters. Rows normalized (across Baden et al. (2016) types) to sum to 1, annotations displaying the number of neurons in percent.

the "**OFF suppressed** 2" type, but also comprised other types, including both OFF and ON cells. Similar to "OFF suppressed 2", cells of "OFF suppressed 1" could be found in most clusters. This is expected, given that both types likely comprise sub-types with distinct functional properties (see coverage factor in Baden et al. (2016)).

We further analyzed why most Baden et al. (2016) types mapped to two or more MDS clusters and why all MDS clusters comprised more than one Baden et al. (2016) type. One hint was that direction- and orientation selective Baden et al. (2016) types (DS and OS in Fig. 2D) were mixed into all three major cluster groups (colored in Fig. 2D), suggesting the clustering did not capture direction selectivity. Consistent with that idea, we found that none of the MDS exhibited significant lateral motion. This is likely because the used digital twin is limited in capturing direction selectivity by both architecture (space-time-separable convolution) and training data (sampling bias in optical flow; Hoefling et al. (2022)). We note that this is a limitation of the predictive model, not of our procedure. Also, the original Baden et al. (2016) types were constructed with additional information on direction selectivity of each cell that our MDS clustering did not have access to. The MDS clustering can therefore not distinguish cells based on their direction selectivity, and instead merged types with and without direction selectivity when they had otherwise similar functional properties. To confirm this, we re-created cell type labels for a fairer comparison using a reduced classifier. This classifier re-labeled cells using only information the MDS clustering also had access to, namely functional responses to "chirp" stimuli, but not direction selectivity or soma size. We found that the reduced classifier performed substantially worse (accuracy dropped from 79% to 49% on test data) and also confused types that were merged by MDS clustering into single clusters (Fig. 2E). For instance, the reduced classifier confused OFF and ON-OFF types that were mapped to MDS clusters 2 and 3, the fast ON types which were mapped mostly to the single MDS cluster 4, and some of the slow ON types mapping to MDS clusters 5 and 6 (Fig. 2D,E). This indicates that without access to direction selectivity or morphological features (e.g. soma size) the finer Baden et al. (2016) types cannot be reliably distinguished. This also explains why our MDS clustering finds fewer clusters.

**Robustness.** To verify that our clustering algorithm returned consistent results across runs, we repeated MDS clustering with another random initialization of the stimulus and initial neuron assignments into 30 instead of five clusters. Again the algorithm converged to seven clusters, which exhibited strikingly similar MDS as the run initialized with five clusters (Appendix Fig. 6). The similarity of cluster assignments across eleven comparison runs varying in initial number of clusters, initial neuron assignment, and MDS initialization was also high (median ARI: 0.67), suggesting MDS clustering is robust across initializations. Further, running MDS clustering for a different digital twin architecture yielded similar results, suggesting robustness to the specific digital twin model used (see Appendix A.6 and Fig. 13 for details).

**MDS provide time-efficient on-the-fly cell type assignments.** As MDS condense the unique response features of each cell type into a single, short stimulus, we next asked if they allow us to assign neurons to types faster than conventional methods. For a fair comparison with conventional approaches, we considered the clusters predicted by the classifier purely trained on the "chirp" responses (see above) as a baseline. For a comparable assessment between clusterings, we merged the 17 "conventional" clusters from the chirp-classifier according to the Baden et al. (2016) hierarchy and matched them to the seven MDS clusters. The baseline classifier retrained on this task predicted the correct cluster label with 73.5% accuracy on a held-out test set of neurons. Recording the required chirp responses (four stimulus repetitions) and assigning functional type labels for new neurons requires 2 min 12 s with this baseline classifier.

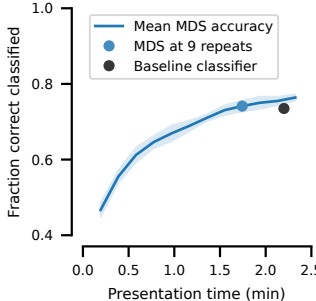

Figure 3: *Accuracy of identifying MDS clusters under simulated observational noise for varying repeats.* Presentation time for the full set of seven MDS. Mean and standard deviation across ten simulations shown. MDS outperform the baseline after 1 min 45 s (9 repeats).

We used the digital twin to simulate an experiment in which we want to obtain the functional type using only MDS. We showed each MDS video of 1.66 s length (50 frames) to each in-silico neuron and assigned them to the functional type of the MDS that elicits the highest simulated response. We simulated multiple MDS repetitions to average out trial-to-trial variability. Repeating the MDS

nine times outperformed the conventional approach in identifying the correct MDS cluster while saving 20% (27 sec) of experimental time compared to the chirp stimulus (Fig. 3). Note that identifying cell types in an experiment requires only the pre-computed MDS. No digital twin training is required to use the MDS as finger-printing stimuli.

**Marmoset retinal ganglion cells.** Next, we demonstrated that our approach is robust across species and recording techniques, using a dataset with the spiking responses of 167 marmoset RGCs to naturalistic videos recorded with multi-electrode arrays. MDS clustering yielded four functional clusters (Fig. 4). Even though the data contained only seven ON cells, the MDS algorithm recovered and even split them into a slow and fast ON cluster (cluster 3 and 4, respectively). OFF cells were also grouped into a fast and slow cluster (cluster 1 and 2, respectively). These clusters likely correspond to ON and OFF midget (slow) and parasol (fast) cells previously described in the primate retina (Field & Chichilnisky, 2007) and are in good agreement with a clustering baseline based on the spike-triggered averages on white noise stimuli (see Appendix A.3 and Fig. 10).

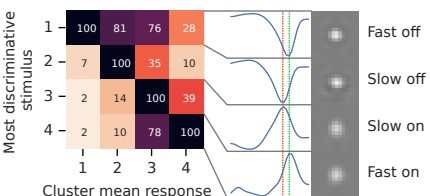

Figure 4: *MDS clustering on marmoset RGCs.* Cluster averaged digital twin response to MDS normalized by mean predictions across the twin's training stimuli. Elements within a column normalized to highest value. MDS space component displayed on symmetric color scale between −1 (black) and 1 (white).

We also verified that repeating MDS clustering with a different seed or initial number of clusters resulted in the same number of final clusters (Appendix Fig. 9). The similarity of clusters across nine comparison runs was also high (median ARI: 0.96). Although the number of cells is smaller and the taxonomy of functional types is less refined for the marmoset retina, the results on this dataset show that MDS clustering finds meaningful clusters across species and recording techniques.

**Macaque visual cortex area V4.** Finally, we showed that our approach extends beyond the retina and also works for brain areas with little domain knowledge about cell types. For that, we tested MDS clustering on spiking responses of neurons in macaque area V4, stimulated by gray-scale images (Willeke et al., 2023). We discarded clusters containing less than six neurons in the test set, yielding 12 well separated MDS clusters (Fig. 5). Here, the MDS were individual images, which showed complex patterns with curvature or texture, typical for V4 (Bashivan et al., 2019; Willeke et al., 2023). Interpreting these MDS suggests that V4 might compute such complex features on the cell type-level. Repeating clustering with a different seed and ten instead of five initial clusters resulted in the same number of MDS clusters (Appendix Fig. 11) with reasonable similarity (ARI: 0.53). Our results suggest that the MDS algorithm can also be successful in computationally more complex brain areas such as primate V4.

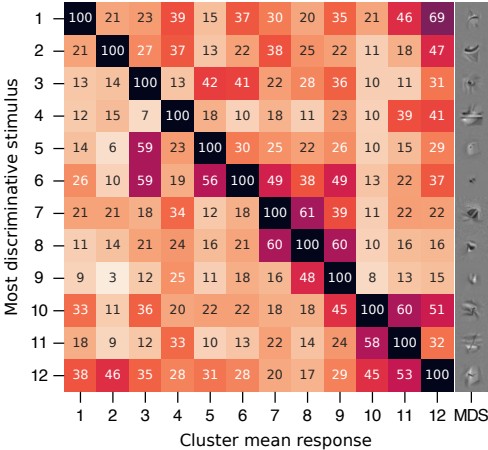

Figure 5: *MDS clustering on macaque V4 on held-out test neurons.* Cluster averaged digital twin response to MDS normalized by mean predictions across the twin's training stimuli. Elements within a column normalized to highest value. MDS displayed on symmetric color scale between -1 (white) and 1 (black).

## 6 DISCUSSION

We presented a novel clustering algorithm that alternates between assigning neurons into functional clusters and optimizing a most discriminative stimulus (MDS) for each functional cluster. We empirically validated our algorithm on three datasets across species, visual processing stages (mouse/marmoset RGCs, macaque V4) and data modalities (gray-scale and multi-channel image and video stimulation, multi-electrode array recordings and calcium-imaging), and showed that MDS allow to identify cell types faster than traditional methods.

**Comparison to other functional clustering methods.** Willeke et al. (2023) reported hints towards functional groups in macaque V4. They optimized a maximally exciting image for each single cell and clustered these MEIs. In contrast, our algorithm computes MDS capturing unique functional properties that distinguish clusters of neurons based on their responses. In mouse V1, Ustyuzhaninov et al. (2022) clustered non-interpretable digital-twin model parameters they assume to represent a neuron's function. In contrast, our algorithm clusters directly based on responses to interpretable MDS, not requiring expert knowledge to extract parameters from the digital twin. Unlike both of these approaches, we empirically validated our clustering algorithm on data where type labels akin to biological ground truth exist (Baden et al., 2016). In addition, once the cell types and their MDS have been established, our approach can identify a neuron's type in new experiments without the digital twin, while both other approaches would need to retrain the digital twin. This suggests that MDS allow for fast online cell typing at recording time, enabling researchers to target a specific cell type they are interested in for data collection and experimental interventions.

**Data-driven and interpretable cell type discovery with MDS.** While previous clustering approaches required domain knowledge to hand-craft fingerprinting stimuli (Farrow & Masland, 2011; Baden et al., 2016; Goetz et al., 2022), our approach identifies functional cell types by automatically generating a fingerprinting stimulus: the MDS. Therefore, our approach is particularly advantageous when domain knowledge is scarce and functional types are yet to be discovered, for example in V1 or – as we demonstrated – in V4: here, without incorporating domain knowledge, our clustering approach successfully identified plausible clusters.

The resulting MDS facilitate interpretability of a cluster's function, as it highlights a cluster's unique visual feature that distinguishes it from the features processed by other groups. For instance, the mouse RGC type "ON contrast suppressed" has been a subject of study for some time (Tien et al., 2015; Baden et al., 2016; Mani & Schwartz, 2017; Tien et al., 2022), but only recently Hoefling et al. (2022) found that this cell type responds strongly to center color opponency – a property that was not revealed by previous fingerprint stimuli, but is clearly visible in our MDS. For complex visual features (e.g. in V4, Fig. 5), describing a cluster's function in words might be more difficult and provides an interesting avenue for future work.

**Limitations and future work.** MDS clustering is applicable for any dataset rich enough to train a digital twin, which is the case for many recent experimental studies (Yamins et al., 2014; Bashivan et al., 2019; Lurz et al., 2020; Sinz et al., 2018; Hoefling et al., 2022; Willeke et al., 2022; Turishcheva et al., 2023; Wang et al., 2023). While a systematic investigation of how the choice of a specific digital twin model affects downstream results is to date unavailable, our MDS and clusters were consistent with another digital twin architecture choice (Appendix A.6 and Fig. 13), suggesting a certain degree of robustness to model details. Another promising direction for future work is to apply our method to new functional recordings where no comparison clustering exists and to validate it with transcriptomic data, e.g. by testing experimentally which cells are driven by our MDS and subsequently identifying their transcriptomic types by PatchSeq (Liu et al., 2020) or spatial transcriptomics (Alon et al., 2021) in the same tissue. While the final judge of the MDS' faithfulness would be an in vivo experiment, single cell MEIs generated with the same optimizer and digital twins as we used were verified in vivo (Willeke et al., 2023) and ex vivo (Hoefling et al., 2022), suggesting that our results might well generalize to an in vivo setting – similar to other work on in vivo verification (Bashivan et al., 2019; Walker et al., 2019; Franke et al., 2022; Tong et al., 2023).

**Algorithm extensions.** ON-OFF RGCs are stimulated by both light increases and decreases. Solely optimizing a single, short MDS may not effectively capture both properties and thus fails to distinguish ON-OFF from pure ON or pure OFF cells. To address this, our method can be expanded to optimize multiple MDS for each cluster, showing promising initial results (see Appendix A.5 and Fig. 12). Additionally, identifying "OFF suppressed 2" cells, characterized by their high baseline firing rate reduced through stimulation, might be improved by searching for a maximally *suppressing* stimulus. As currently set up, MDS clustering assumes discrete cell types, and future work could extend it towards accounting for recently reported continuity within or between cell types (Tasic et al., 2016; Harris et al., 2018; Tasic et al., 2018; Stanley et al., 2020; Scala et al., 2021).

**Conclusions.** Our general-purpose functional cell type clustering algorithm and clusters' most discriminative stimuli could be a useful tool for the neuroscience community and spark further work on exploring functional groups across the visual system – especially in areas where no fingerprinting stimuli are available – and help designing experiments previously limited by experiment time.

## REPRODUCIBILITY STATEMENT

We will make our code, seeds, and used Python environment publicly available for reproducibility at https://github.com/ecker-lab/most-discriminative-stimuli. Furthermore, we provide detailed method descriptions and parameters in sections Most discriminative stimulus clustering algorithm, Data and implementation details, and Appendix. For all results we quantified reproducibility of MDS clustering across runs with varying parameter initializations and we provide according visualizations in Figs. 6, 9 and 11.

## ACKNOWLEDGMENTS

The authors would like to thank (in alphabetic order): Catie Nealley, Emmanouil Froudarakis, Federico D'Agostino, Gabrielle Rodriguez, Jiakun Fu, Katrin Franke, Kayla Ponder, Polina Turishcheva, Saumil Patel, Taliah Muhammad, Tori Shinn. MFB, KFW, and SM thank the International Max Planck Research School for Intelligent Systems (IMPRS-IS). This project has received funding from the European Research Council (ERC) under the European Union's Horizon Europe research and innovation program (Grant agreement No. 101041669). This publication was funded by the Deutsche Forschungsgemeinschaft (DFG, German Research Foundation) - Project-ID 432680300 - SFB 1456. This work was supported by the German Research Foundation (DFG): SFB 1233, Robust Vision: Inference Principles and Neural Mechanisms, TP4, project number: 276693517. MB and PB are members of the Machine Learning Cluster of Excellence "Machine Learning — New Perspectives for Science", funded by the Deutsche Forschungsgemeinschaft (DFG, German Research Foundation) under Germany's Excellence Strategy – EXC number 2064/1 – Project number 390727645. We thank the European Union (ERC, "NextMechMod", ref. 101039115). Views and opinions expressed are however those of the authors only and do not necessarily reflect those of the European Union or the European Research Council Executive Agency. Neither the European Union nor the granting authority can be held responsible for them. We also thank the Hertie Foundation for support. This work was supported by the Tübingen AI Center. The work was supported by the Intelligence Advanced Research Projects Activity (IARPA) via Department of Interior/ Interior Business Center (DoI/IBC) contract number D16PC00003. The U.S. Government is authorized to reproduce and distribute reprints for Governmental purposes notwithstanding any copyright annotation thereon. AST acknowledges support from NSF NeuroNex grant 1707400. AST acknowledges support from National Institute of Mental Health and National Institute of Neurological Disorders And Stroke under Award Number U19MH114830 and National Eye Institute award numbers R01 EY026927 and Core Grant for Vision Research T32-EY-002520-37. Disclaimer: The views and conclusions contained herein are those of the authors and should not be interpreted as necessarily representing the official policies or endorsements, either expressed or implied, of IARPA, DoI/IBC, or the U.S. Government. The authors gratefully acknowledge the computing time made available to them on the high-performance computers HLRN-IV at GWDG at the NHR Center NHR@Göttingen.

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

# A   APPENDIX

## A.1   SUB-CLUSTER OPTIMIZATION DETAILS

To optimize MDS for sub-cluster candidates, we needed to avoid the trivial case where all neurons are in one cluster with the same MDS as before splitting. Thus, we first optimized MDS for the sub-clusters only and reassigned neurons after each stimulus optimization step. Hence, in the beginning of this optimization, neuron reassignments were done on noisy stimuli to break symmetry. After iterating this procedure for 50 steps, we performed one global optimization step considering all original clusters and the newly created sub-cluster candidates. To do so, we kept neurons' assignments to the clusters and randomly initialized the according MDS. We then optimized the MDS for all clusters and reassigned neurons based on them. Now, the neuron assignments and MDS accurately reflect the new optimal clustering state. Then, we evaluated the mean objective across all clusters ($\langle J_c \rangle_c$, Eq. (1)) and kept the new sub-clusters if they improved overall clustering.

## A.2   MOUSE RETINA DETAILS

**Robustness check.** As a simple robustness check of MDS clustering, we ran clustering with 30 instead of 5 initial clusters. The resulting clusters and average responses are visualized in Fig. 6. This run had a rather high adjusted rand index of 0.87, thus, we asked how similar the result shown in Fig. 2 would be compared to a typical run. Therefore, out of eleven comparison runs varying in initial number of clusters, initial neuron assignment, and MDS initialization, we compared to the run with median adjusted rand index (ARI: 0.67) and found overall good agreement (compare Fig. 7 to Fig. 2).

**Response traces.** To analyze individual clusters it might be useful to inspect response traces of neurons. Fig. 8 shows mean responses for each MDS and cluster combination, for which response traces are grouped by Baden et al. (2016) RGC types.

**Digital twin.** The digital twin of the mouse retina was an ensemble of five identically structured convolutional neural networks (CNN) trained to predict inferred firing rates (from calcium imaging) of RGCs in response to dichromatic natural movies previously published by Hoefling et al. (2022). In this ensemble, each member consisted of a core module, that was shared between all neurons, and a neuron specific readout module Klindt et al. (2017). The core was implemented as a CNN with two convolutional layers, each with 16 features. Both convolutional layers consisted of space-time separable 3D convolutional kernels, batch normalization layer and an ELU nonlinearity (Clevert et al., 2015). In the first layer, sixteen $2 \times 11 \times 11 \times 21$ (channels × height × width × frames) kernels were applied as valid convolution; in the second layer, sixteen $16 \times 5 \times 5 \times 11$ kernels were applied with zero padding along the spatial dimensions. The temporal kernels were parameterized as Fourier series. To account for inter-experimental variability affecting the speed of retinal processing, the model included a time stretching parameter trained for each recording separately Zhao et al. (2020). The readout module modeled the spatial receptive field (RF) of each neuron as a 2D isotropic Gaussian, parameterized as $\mathcal{N}(\mu_x, \mu_y; \sigma)$. The model's output, i.e. the predicted RGC responses, were implemented as an affine function of the feature maps of the core module, weighted by the spatial RF from the readout module, followed by a softplus nonlinearity. During inference, predicted responses were averaged over all five members of the ensemble.

**Digital twin training.** Each member was initialized with a different seed and trained independently using the Adam optimizer (Kingma & Ba, 2015) minimizing a Poisson loss,

$$\mathcal{L}_{\text{Poisson}} = \sum_{n=1}^{N} \left( \hat{r}^{(n)} - r^{(n)} \ln \hat{r}^{(n)} \right) , \qquad (2)$$

where $N$ is the number of neurons, and $r^{(n)}$ and $\hat{r}^{(n)}$ are the measured and predicted firing rate of a neuron $n$ for an input of duration of $t = 50$ frames, respectively. The batch size was set to 32 and a chunk size (number of frames for each element in the batch) to 50.

The training schedule was based on Lurz et al. (2020) and used early stopping (Prechelt, 2002) based on the validation set, consisting of 15 out of the 108 movie clips. If the mean correlation between predicted and measured neuronal responses failed to increase on the validation set during any five consecutive passes through the entire training set (epochs), the training was stopped and the model

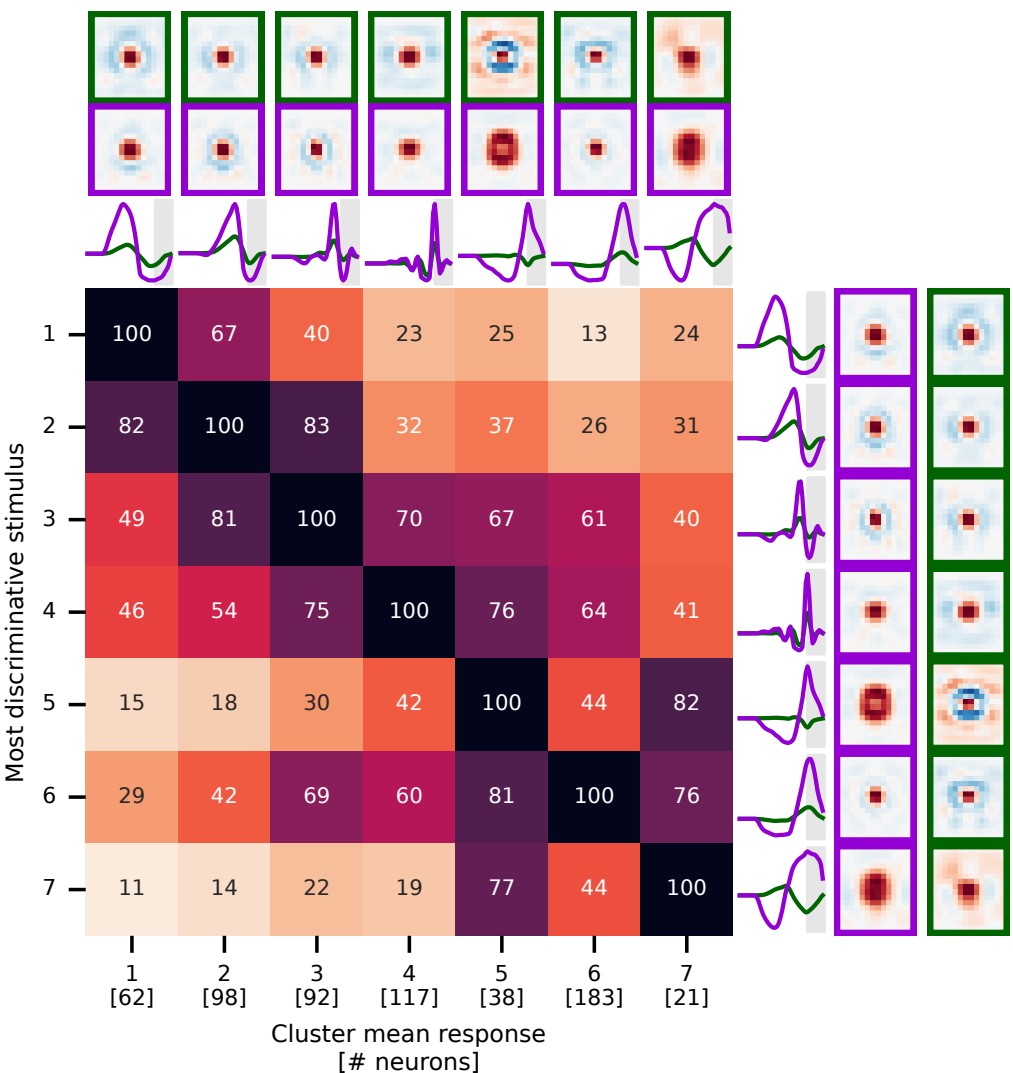

Figure 6: Mouse retina results of clustering run with 30 instead of 5 initial clusters with another random initialization of stimulus and neuron assignments. The final number of clusters is lower than 30, as empty clusters are removed while running MDS clustering.

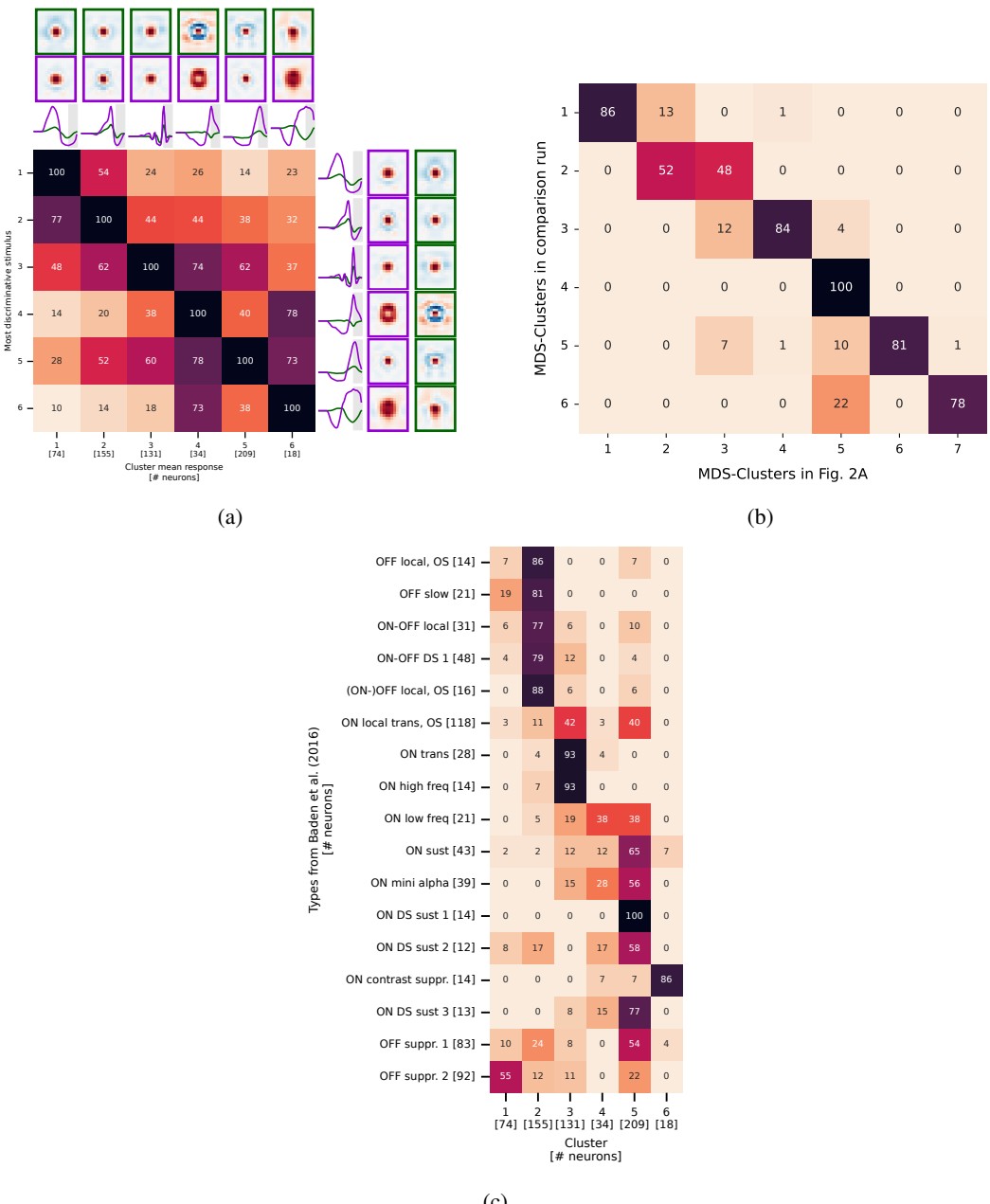

Figure 7: Results of a typical clustering run (run with median ARI compared to results in Fig. 2). Results on held-out test neurons and empirical comparison to Baden et al. (2016) types shown. **A.** Cluster averaged digital twin response to the optimized MDS. Elements within a column normalized to highest value. **B.** Confusion matrix between neuron's MDS cluster assignment in Fig. 2 and the most typical comparison run. **C.** Confusion matrix between MDS clusters and Baden et al. (2016) types. Elements within rows were normalized to sum to 1, annotated numbers display the number of neurons in percent.

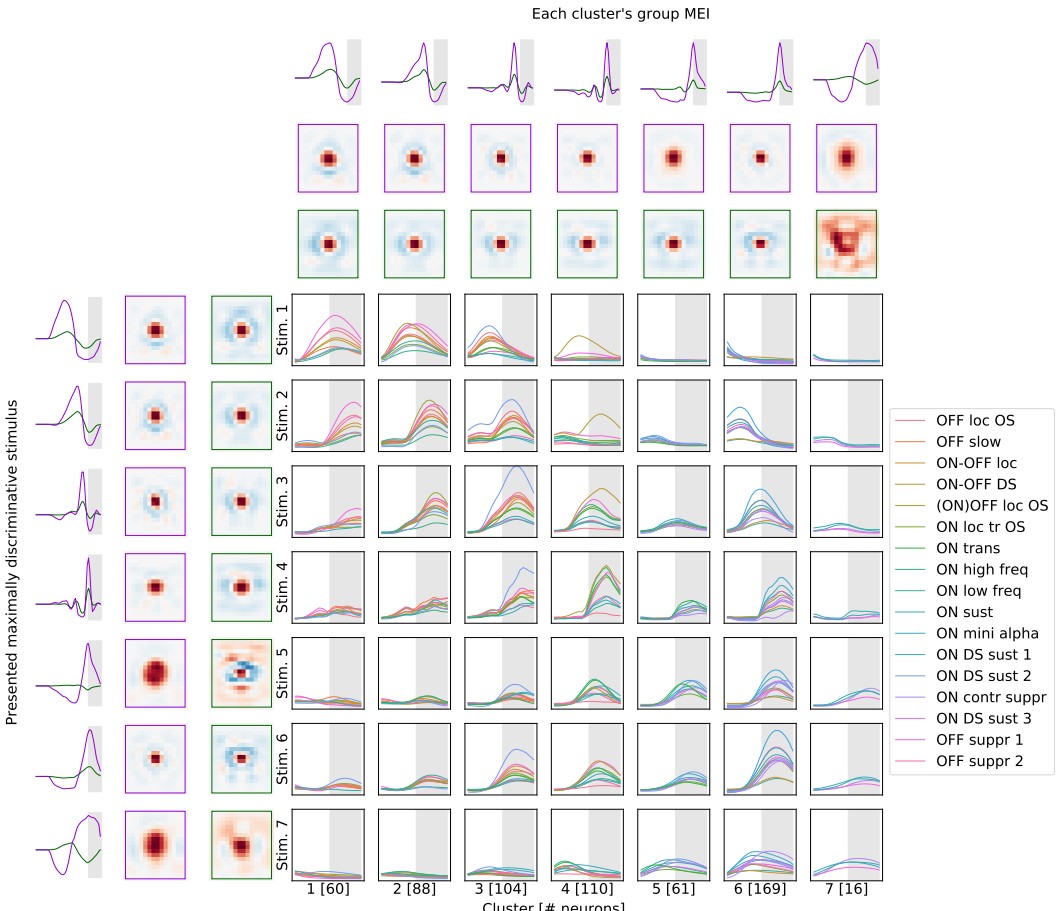

Figure 8: Mean traces of held-out test neurons in response to the most discriminative stimuli (left) grouped by the Baden et al. (2016) RGC type. Group MEIs for each cluster are visualized on top of the plot. The extend of the x- and y-axis is shared across all plots with response traces.

checkpoint performing best on the validation set was restored. This process of early stopping and weight restoring was repeated four times, each time reducing the initial learning rate of 0.01 by a learning rate decay factor of 0.3.

### A.3 MARMOSET RETINA DETAILS

**Robustness check.** As a simple robustness check of MDS clustering, we compared to a typical comparison run (the one with mean ARI compared to Fig. 4 selected across nine comparison runs varying in the number of initial clusters, intial cluster assignment, and MDS initialization). The resulting clusters and average responses are visualized in Fig. 9.

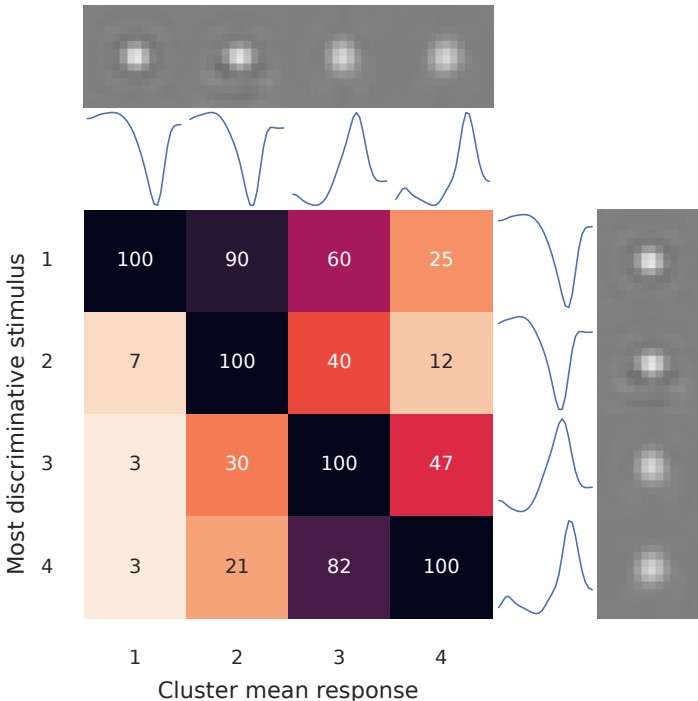

Figure 9: Marmoset retina results of typical clustering run with five instead of ten initial clusters with a different random initialization of stimulus and neuron assignments. Responses normalized by mean of digital twin predictions across the twin's training set.

**Digital twin.** The model trained to predict marmoset RGC responses was a CNN with a core and readout structure similar to Hoefling et al. (2022). The core consisted of 4 layers with space-time separated kernels of shape $21 \times 21$ pixels (height $\times$ width) for the spatial kernel and 27 frames for the temporal kernel in the first layer. In the following layers, the spatial kernel was of $5 \times 5$ pixels size and the temporal kernel covered 5 frames. The number of channels increased from 8 in the first layer to 16, 32 and 64 in subsequent layers. The kernels of the first layer were smoothed adding a 2D Laplace filter in the spatial dimensions and a 1D Laplace in the temporal dimensions as a prior. In contrast to Hoefling et al. (2022) temporal kernels were not parameterized as Fourier series. In the readout, each cell's receptive field (RF) was modeled as an isotropic Gaussian. The response function was then modeled as an affine function of the core's weighted feature maps at the RF positions, followed by a softplus. The feature map weight vector was regularized using L1-norm to enforce sparsity.

**Digital twin training.** We trained the CNN using the Adam optimizer (Kingma & Ba, 2015) optimizing the Poisson loss (Eq. (2)) on 16 out of 20 available 5 minute trials. The batch size was 16 and we used early stopping on the correlation between predicted and recorded responses on the validation set which consisted of the 4 remaining trials. We reduced the learning rate by a factor

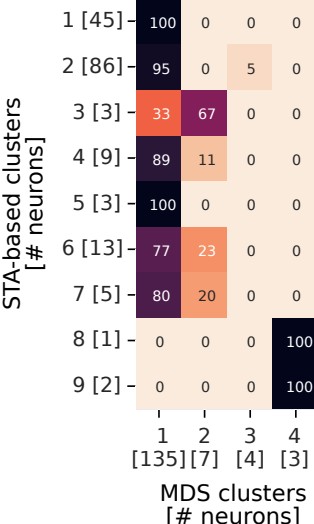

Figure 10: Confusion matrix to STA-based clustering provided with the marmoset data. Elements within rows normalized to sum to 1, annotated number of RGCs in percent.

of 0.1 each time the validation correlation did not improve for 5 epochs. The initial learning rate was 0.008 and the minimal learning rate we allowed was $8 \cdot 10^{-5}$. We stopped training early if the validation correlation did not improve for 50 epochs.

**MDS optimization.** To optimize the MDS on this dataset we used an ensemble of 5 of the above described models varying in their initialization before model training. We also tailored a few of the parameters of MDS optimization to this dataset, optimizing 39 frames $\times$ $(80 \times 90)$ pixels one-channel MDS videos (85 Hz frame-rate) with a learning rate of 4 and chose ten initial clusters. Unlike for the mouse retina data, here we fed the predicted activity of only the last time-bin to the discriminative objective function (Eq. (1)). The MDS input was constraint with an $L_2$-norm of 3.5 and the pixel values were clipped to stay within the range of -1 and 1. We optimized the stimulus for a maximum of 10,000 optimization steps, and for 10 steps during splitting, or until an exponential moving average of the objective (Eq. (1)) did not improve for 100 optimization steps. During splitting, we created two new sub-cluster candidates and terminated splitting after 100 iterations. As neuron's responses are not normalized in this data, we post-hoc normalized digital twin predictions such that for each neuron, predictions were standard-normal distributed across the digital twin's training stimuli, ensuring single cell responses do not dominate the within-cluster mean.

**STA-based cell clustering.** To be able to compare our MDS clustering of marmoset RGCs to an alternative clustering approach, we determined each cell's RF size as the area within the 1.5 standard deviation contour of a Gaussian fitted to the cell's spatial component of the spike triggered average (STA) measured under spatio-temporal white-noise stimulation. The STA's temporal component and the RF size were taken as a feature vector for each cell. The feature vectors were normalized and their dimensionality was reduced using Principal Component Analysis (PCA). We selected as many principal components as necessary to explain 90% of the variance. We then used KMeans++ (Arthur & Vassilvitskii, 2007) on the reduced-dimensionality feature vectors to determine 4-6 initial clusters. These clusters were then manually curated to ensure each cluster had similarly-sized cells and tiled the retinal surface, resulting in a total of nine clusters. However, the taxonomy of functional types in the marmoset is less understood compared to the mouse and a concise interpretation of these clusters is not yet established, so they cannot be treated as ground truth. Comparing these clusters to our MDS approach yielded overall good agreement (Fig. 10).

**Data recording.** All marmoset data were obtained by recording RGC spikes extracellularly from isolated retina placed on a MEA, as explained in Krüppel et al. (2023). Marmoset retinal tis-

sue was obtained immediately after euthanasia from animals used by other researchers, in accordance with national and institutional guidelines and as approved by the institutional animal care committee of the German Primate Center and by the responsible regional government office (Niedersächsisches Landesamt für Verbraucherschutz und Lebensmittelsicherheit, permit number 33.19-42502-04-20/3458).

### A.4 MACAQUE VISUAL AREA V4 DETAILS

**Robustness check.** As a simple robustness check of MDS clustering, we ran clustering with 10 instead of 5 initial clusters. The resulting clusters and average responses are visualized in Fig. 11.

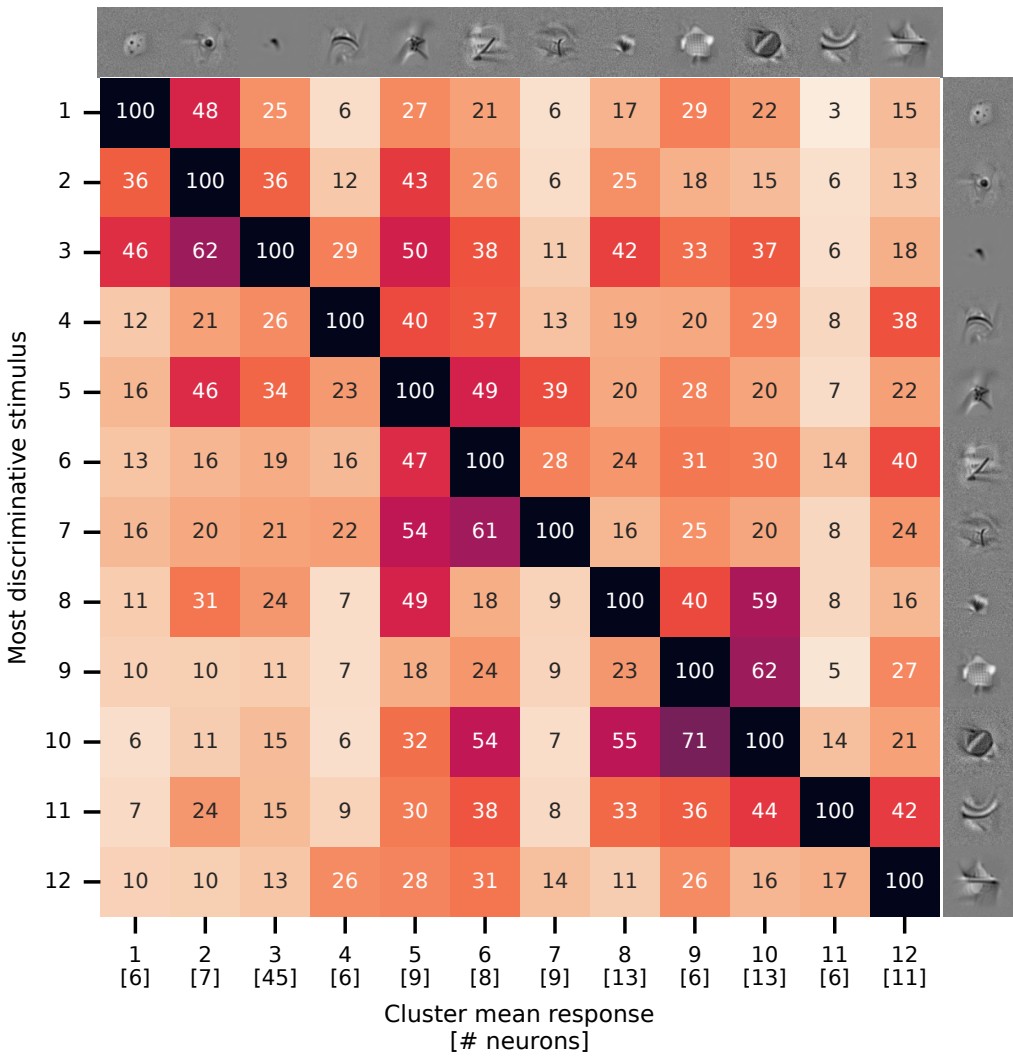

Figure 11: Macaque V4 results of clustering run with 10 instead of 5 initial clusters with another random initialization of stimulus and neuron assignments. Responses normalized by mean of digital twin predictions across the twin's training set.

**Digital twin.** The digital twin of macaque V4 neurons, published by Willeke et al. (2023), again followed the core-readout structure. The core of the model consisted of a ResNet50 (He et al., 2016) which was adversarially trained on ImageNet (Deng et al., 2009) to have robust visual representations (Salman et al., 2020). In the digital twin's core, the first residual block of layer 3 (layer-3.0) of the ResNet trained with image perturbation constraint $\epsilon = 0.1$ was selected to read out from, be-

cause it yielded the highest predictive performance, compared to all other ResNet models and layers. The corresponding size of the output feature map was 1024. The model used batch-normalization (Ioffe & Szegedy, 2015). Lastly, the resulting tensor was rectified with a ReLU unit to obtain the final nonlinear feature space. To predict the response of a single neuron from the feature space, a *Gaussian readout* was used (Lurz et al., 2020). For each neuron $n$, this readout learned spatial coordinates of the position of the receptive field on the output tensor and extracted a feature vector of length *channels* at this location. This extracted feature vector was then used in a linear-nonlinear model to predict the neuronal response. To this end, an affine function of the resulting feature vector at the chosen location was computed, followed by a rectifying nonlinearity, chosen to be an ELU (Clevert et al., 2015) offset by one ($ELU(x) + 1$) to make responses positive. The weight vector was $L_1$-regularized during training.

**Digital twin training.** The model was trained to minimize the Poisson loss (Eq. (2)) summed across neurons and computed between observed spike counts and the models' predicted spike counts with an additional $L_1$ regularization of the readout parameters. Models were trained on the full dataset of $n = 100$ recording sessions with $n = 1244$ neurons and an image size of 100 by 100 pixels. After each epoch, the Poisson loss was computed on the entire validation set. Early stopping was used as follows: If the validation loss did not improve for five consecutive epochs, we restored the weights with the lowest Poisson loss on the validation set and down-scaled the learning rate by a factor of $0.3$ before training continued. After four early stopping steps were completed, the training was stopped.

**MDS optimization.** We tailored a few MDS optimization parameters to the V4 dataset, optimizing single gray-scale ($100 \times 100$) pixel images with a learning rate of 20, before each optimization step we smoothed image gradients by convolving a Gaussian filter with standard deviation 2 and after each optimization step we normalized the image to a $L_2$-norm of 35. We optimized MDS for 666 steps, and for 1 step during splitting. We stopped MDS optimization early if the exponential moving average of the objective (Eq. (1)) did not improve for 100 optimization steps. Furthermore, during splitting we created two new sub-cluster candidates, and terminated splitting after 2 iterations when initializing with 10 clusters and 3 iterations when initializing with 5 clusters (theoretically allowing for the creation of 40 clusters each). As neurons' responses are not normalized in this data, we post-hoc normalized digital twin predictions such that for each neuron predictions were standard-normal distributed across the digital twin's training stimuli, ensuring single cell responses do not dominate the within-cluster mean.

## A.5 EXTENSION TO TWO MDS PER CLUSTER

To investigate if our method could be extended to increase cluster granularity by optimizing multiple stimuli, we performed a proof-of-concept experiment: we optimized two MDS per cluster. The responses to both stimuli need to be aggregated before feeding them to the objective (Eq. (1)). Here, we chose the product of the responses across stimuli, a soft and differentiable implementation of the "logical and" operation that requires neurons to respond to both stimuli. Initializing with five randomly assigned clusters for the mouse retina, we found a similar result as when optimizing only one stimulus, with a crucial difference: cluster three now showed two different MDS with a sign-flipped temporal UV dependence (Fig. 12A). Only ON-OFF cells would respond to both of these stimuli, and consequently the neurons of this new cluster were mainly associated with ON-OFF Baden et al. (2016) types (Fig. 12B). Interestingly, the MDS also showed orientation-selective features, and hence the according cluster also contained orientation-selective Baden et al. (2016) types.

While this proof-of-concept is very promising, the extension to multiple stimuli raises a number of new questions. For instance, there are several ways of aggregating responses from multiple stimuli (for example taking the mean, the minimum, the maximum, etc.). Further, optimizing more than two stimuli could be beneficial, and adding a diversity term between stimuli (Cadena et al., 2018; Ding et al., 2023) could improve clustering. Such extensive expansions are beyond the scope of the current paper, and we propose them as a fruitful avenue to further improve optimization-based functional clustering.

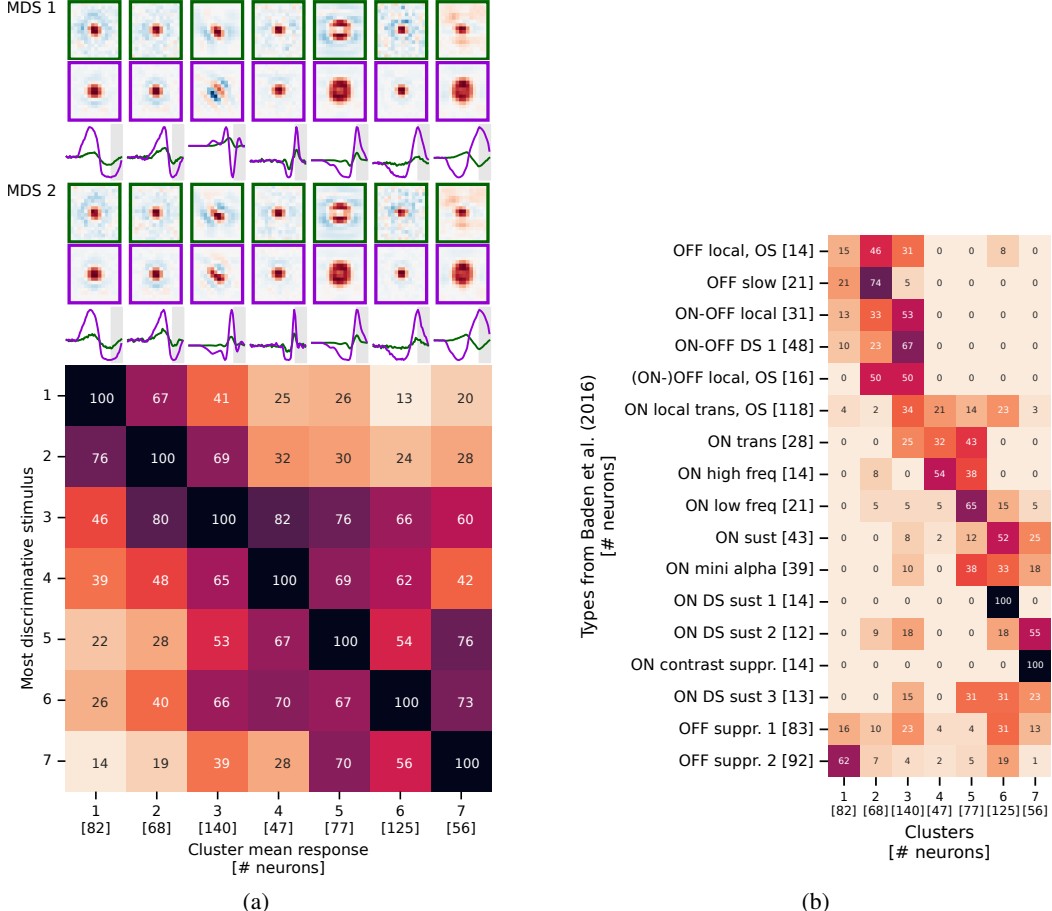

(a)                                                                                     (b)

Figure 12: *Proof of concept of clustering based on two MDS on mouse RGCs.* Results on held-out test neurons shown. **A.** Cluster averaged digital twin response to the optimized MDS. To obtain the response for a cell, the product of the responses to each individual MDS is computed. Elements within a column normalized to highest value. **B.** Confusion matrix between MDS clusters and Baden et al. (2016) types. Elements within rows were normalized to sum to 1, annotated numbers display the number of neurons in percent.

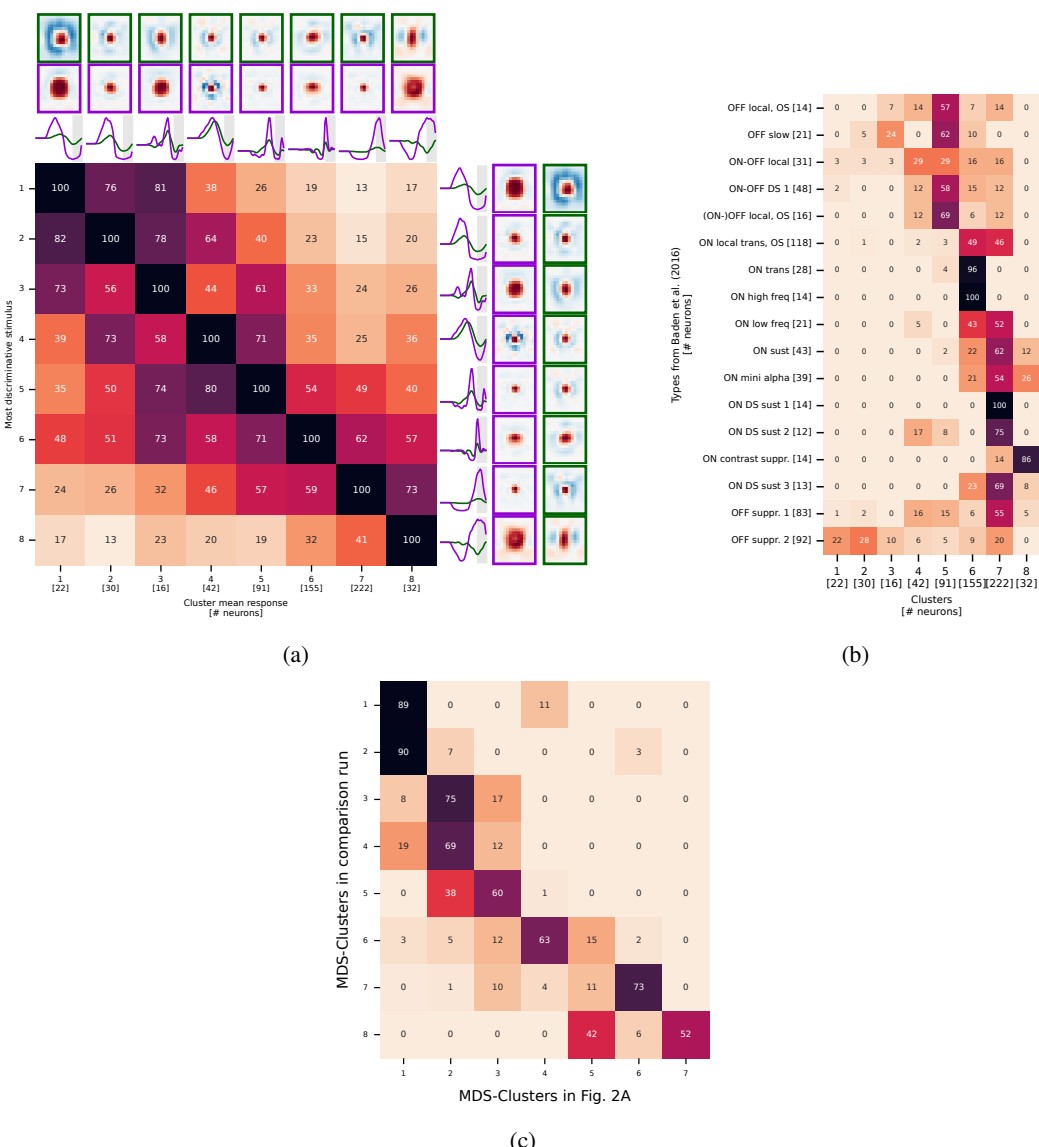

(a)

(b)

(c)

Figure 13: *Most discriminative stimuli clustering on mouse RGCs for another digital twin model architecture.* Results on held-out test neurons and empirical comparison to Baden et al. (2016) types shown. **A.** Cluster averaged digital twin response to the optimized MDS. Elements within a column normalized to highest value. **B.** Confusion matrix between MDS clusters and Baden et al. (2016) types. Elements within rows were normalized to sum to 1, annotated numbers display the number of neurons in percent. **C.** Confusion matrix between the MDS cluster assignments for the modified digital twin architecture and MDS cluster assignments reported for the original digital twin used for Fig. 2. Elements within rows were normalized to sum to 1, annotated numbers display the number of neurons in percent.

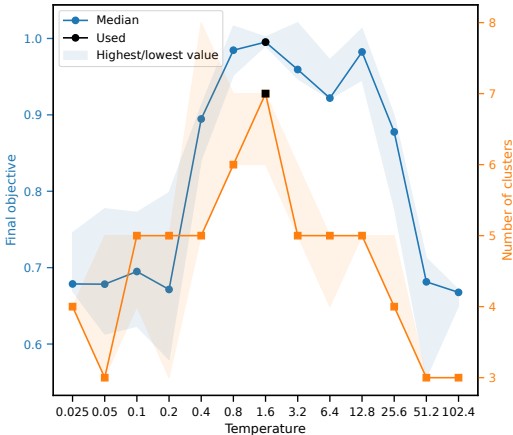

Figure 14: *Mouse retina clustering results vs. objective temperature.* For each temperature value, we optimized three runs varying in stimulus and cluster assignment initialization.

### A.6 RESULTS FOR ANOTHER MOUSE RETINA DIGITAL TWIN ARCHITECTURE

To investigate how the choice of a digital twin architecture affects the clustering results, we modified the mouse retina digital twin model (Appendix A.2) by adding a third layer and varying the size of the convolutional kernels. Specifically, we changed the kernels of the second layer to sixteen $8 \times 3 \times 3 \times 9$ (channels $\times$ height $\times$ width $\times$ frames) kernels, and introduced a third layer of sixteen $16 \times 3 \times 3 \times 3$ kernels, both applied with zero padding along the spatial dimensions. We trained five of these digital twins initialized with different random seeds and used the same training schedule as Hoefling et al. (2022) (see our Appendix A.2 for details). We applied our MDS clustering to this ensemble of digital twins. The resulting clusters and MDS looked similar for both digital twin neural network architectures (compare Fig. 2 to Fig. 13), suggesting that while details may differ across digital twin models, the overall result remains the same.

### A.7 ANALYSIS OF TEMPERATURE HYPER-PARAMETER

We asked how the choice of a specific temperature would affect MDS clustering. For the mouse retina, we systematically swept temperatures logarithmically and ran MDS clustering for three different initializations. We found a broad, flat objective optimum for temperature values between 0.8 and 12.8 with a consistent number of five to seven resulting clusters (Fig. 14). In general, all clustering algorithms will slightly over- or under-cluster, and to determine a precise number of cell types, functional clusters need to be verified by comparison with morphological, genetic, or transcriptomic data. The analyses in our paper are based on temperature 1.6 with the highest objective value, resulting in seven clusters that we successfully matched to an existing hierarchy of cell types (Baden et al., 2016).

For lower or higher temperatures, the objective value and the number of clusters we found decreased, indicating worse clustering performance. This results directly from the structure of our softmax objective function (Eq. (1)). For high temperatures, $\tau \gg \bar{r}_k(x_c)$, $k = 1, 2, ..., c, ...$, the inputs to the objective $\bar{r}_k(x_c)/\tau$ are dominated by the temperature, $\tau$. Hence, the objective becomes insensitive to stimulus-dependent inputs $\bar{r}_k(x_c)$, and clustering performance decreases. For small temperatures, $\tau \ll \bar{r}_k(x_c)$, the inputs will be strongly amplified and the response of a single cluster $\bar{r}_{k^*}(x_c)$ will dominate the objective, making it insensitive to the inputs of all other clusters $\bar{r}_{k \neq k^*}(x_c)$. The resulting MDS will then not suppress all clusters. In conclusion, a good temperature balances these two undesirable regimes.

