# OpenReview forum: "Most discriminative stimuli for functional cell type clustering"
_ICLR.cc/2024/Conference — ICLR 2024 poster_

### Official Review · Reviewer_NLuU · 2023-10-18

**Soundness:** 2 fair
**Presentation:** 2 fair
**Contribution:** 3 good
**Rating:** 5
**Confidence:** 2

**Summary:**

An optimisation-based clustering approach is proposed obtain functional clusters of neurons using deep predictive models and Maximally Discriminative Stimuli (MDS). The stimulus optimization and cluster reassignment are alternately conducted until the convergence is achieved. Empirical studies demonstrate the proposed algorithm recovers functional clusters in mouse retina, marmoset retina and macaque visual area V4.

**Strengths:**

++ The paper attempts to address an important problem of cell type identification.

++ The experimental results seem to be promising and useful in practice.

**Weaknesses:**

-- Essential technical details of the digital twin (i.e. the neural network used to predict the neuron responses) are missing. From the text, it is not clear how the digital twin is obtained and how it will affect the optimisation of MDS.

-- It is stated in the paper "We optimized the stimulus by gradient-ascent, using the SGD optimizer with a learning rate of 100". What are the learnable parameters during this optmization? It is unusual to use such a large value of learning rate.

-- The statement is ambiguous "After each step we renormalize the video snippet to a L2 norm of 30 and clamp ... into ranges of [-0.913, 6.269] and [-0.654, 6.269]". What does it mean by "a L2 norm of 30"? What are the ranges from?

**Questions:**

1. Where does the digital twin (i.e. the neural network) come from? Is it a pre-trained neural network?
2. How are the results/performance sensitive to the hyper-parameters and initializations?

---

> ### Author Response · Authors · 2023-11-12
>
> Thank you for your careful assessment of our paper and the positive feedback you mentioned as strengths. You raise three concerns related to (a) details on the model, (b) learning rate for stimulus optimization and (c) stimulus normalization details, and two questions (1. + 2.), which we are happy to clarify below. As these concerns seemed rather minor and related to presentation, we were somewhat surprised by the low score. Please let us know whether our following answers address your concern and consider updating your score accordingly. If you have remaining concerns, please let us know what we should do to address them.
>
> - a. The models are convolutional neural networks in the well-established core-readout architecture (Klindt et al., NeurIPS 2017; Lurz et al., ICLR 2021) trained from scratch (no pre-training) on neural responses to images/videos. The detailed parameters vary from dataset to dataset and have been optimized by the authors of previous papers to maximize the predictive power of the models on the respective datasets (mouse retina: Höfling et al. bioRxiv 2022; macaque V4: Willeke et al, bioRxiv 2023; marmoset retina: work soon to be submitted). We will include an appendix that describes the technical details of the models.
>
> - b. The learnable parameters are the pixel values of the maximally discriminable images/video segments, while the network weights are treated as fixed and not being optimized. The high learning rate is typical for pre-image search problems like activity maximization and has been selected as the best one from within the range we explored (10^-3 to 10^3). We will add this information to the paper.
>
> - c. The normalization is done because neural responses usually increase with the contrast of the input, but the monitor that displayed the digital twin’s training videos has a finite dynamic range. If we had not constrained the overall energy of the stimuli, the optimized MDS would show an unrealistically high contrast that cannot be displayed by the monitor. This procedure is typical in activity maximization settings (e.g. Walker et al., Nature Neuroscience 2019; Hoefling et al., bioRxiv 2022; Willeke et al, bioRxiv 2023). Even with constraining the L2 norm, single pixels can still exceed the monitor range, which is why we clipped them. The provided intervals correspond to the minimum (UV: -0.913, green: -0.654) and maximum (UV and green: 6.269) light intensity of the monitor, respectively, whereas zero corresponds to the mean luminance across the stimuli. The intervals are asymmetric, because the distribution of light intensities in the stimuli is skewed. We will add this additional information to the paper.
>
> 1. The digital twin models were trained from scratch on neural responses by authors of previous papers. See answer to a.
>
> 2. The three hyperparameters of the method are learning rate, number of initial clusters, and temperature (Eq. 1). The learning rate affects primarily the runtime of the optimization procedure, not the resulting optimal stimuli. We verified that the results are robust with respect to different initializations and number of initial clusters (mouse retina: adjusted rand index ARI=0.87 (page 7) and compare Figure 2A to Appendix Figure 7; marmoset retina: ARI=0.49 (p. 8) and compare Figure 4A to Appendix Figure 8; macaque V4: ARI=0.53 (p. 8) and compare Figure 5 to Appendix Figure 9). The temperature determines how strongly we penalize larger cluster responses. We explored various temperatures in early tests and 1.6 worked well, and we will now run a control where we examine it more systematically and report the results.

---

> > ### Author Response · Authors · 2023-11-17
> >
> > Regarding your question on how our method is sensitive to the temperature hyper-parameter, we now can expand on our previous answer. We systematically swept temperatures logarithmically and ran MDS clustering for three different initializations, and report the median objectives in the table below (for the mouse retina data).
> >
> > | Temperature |     Objective (Median) |
> > |------------:|-----------:|
> > |   0.2       |  0.67      |
> > |   0.4       |  0.89      |
> > |   **0.8**   |  **0.98**  |
> > |   **1.6**   |  **0.99**  |
> > |   **3.2**   |  **0.95**  |
> > |   **6.4**   |  **0.92**  |
> > |  **12.8**   |  **0.98**  |
> > |  25.6       |  0.87      |
> > |  51.2       |  0.68      |
> >
> >
> > We found a broad, flat objective optimum (bold-face numbers) for temperature values between 0.8 and 12.8 with a consistent number of five to seven resulting clusters. Generally, all clustering algorithms will slightly over- or under-cluster, and to determine a precise number of cell types, functional clusters need to be verified by comparing to morphological, genetical, or transcriptomics data. Analysis in our paper are based on temperature 1.6 with the highest objective value resulting in seven clusters that we successfully matched to an existing hierarchy of cell types (Baden et al. 2016).
> >
> > For lower/higher temperatures, the objective value and the number of clusters found decreased, indicating worse clustering performance. This results directly from the structure of our softmax objective function,
> >
> > $$
> >     \max_{x_c}J_c=\max_{x_c} \left( \log \frac{\exp \left(\bar r_c(x_c) / \tau \right)}{ \frac{1}{K}\sum_{k=1}^K \exp \left(\bar r_k(x_c) / \tau\right)} \right) \quad \text{(Eq. 1)}\ .
> > $$
> >
> > For high temperatures, $\tau \gg \bar r_k(x_c),\ k=1, 2, ..., c, ...$, the inputs to the objective $\bar r_k(x_c) / \tau$ are dominated by the temperature, $\tau$. Hence, the objective becomes insensitive to stimulus dependent inputs $\bar r_k(x_c)$, and clustering performance diminishes. For small temperatures, $\tau \ll \bar r_k(x_c)$, inputs will be strongly amplified and the response of a single cluster $\bar r_{k^*}(x_c)$ will dominate the objective, making it insensitive to the inputs of all other clusters $\bar r_{k \neq k^*}(x_c)$. The resulting MDS will then not suppress all clusters. In conclusion, a good temperature balances these two undesirable regimes.
> >
> > We will add results of this analysis as a figure and a discussion to the Appendix of the final version of the paper. Thank you very much for asking this interesting question, which will make our paper stronger.

---

> ### Comment · Reviewer_NLuU · 2023-11-21
> **Why split cells into a set of 80% for clustering and a held-out test set?**
>
> I appreciate the authors' response to my comments. The clarifications help a lot to better understand the work. I'm still confused in the experiment setting "we split cells into a set of 80% used during clustering and report results on the remaining held-out test set".  Since the method is an unsupervised learning algorithm, why not directly apply it to the full set of neurons and report the results of clustering? How would it be used in a practical application when there are no reference/ground truth functional types? Can the data used for training the digital twin be used for such functional type clustering? E.g., clustering directly on the responses of the training data (used for digital twin network training)?
> The essential prior knowledge and motivations of this work are missing and I hope the revised version can be significantly improved in such aspects for better readability.

---

> > ### Author Response · Authors · 2023-11-22
> >
> > The split into training and test set is a safeguard against overfitting, required (e.g.) for an unbiased estimate of the cross-response matrix (Figs. 2A, 4, 5). We want the stimuli obtained by our procedure to also work in future experiments, where we do not show natural stimuli and do not train a digital twin, but show only the MDS. To assess how well this is expected to work, we exclude a fraction of the neurons from training to make sure the images are not overfit to the training set. You are of course right that for assessing how well the clusters we obtain correspond to ground truth labels (Fig. 2D), we could have used the training set, too. We found it more straightforward to do all analyses only on the test set for consistency, but if you think that we should do the comparison of cluster assignments to ground truth labels on the full dataset, we can easily include that, too.
> >
> > Why do we not cluster on the responses of the training data? The receptive field locations of neurons vary. Therefore, each neuron “saw” a different stimulus during training. Hence, even two neurons with identical input-output function will not have the same response to natural stimuli. As a consequence, we cannot simply cluster the responses. Thanks for pointing out that some of the basic motivations that are obvious to us are not explained well enough yet. We will do another iteration on the introduction and motivation for the final version.
> >
> > In summary, the revision we just posted now clarifies your question about details on the digital twin and reports several (new) experiments on the robustness w.r.t. hyperparameters:
> > 1. The procedure is robust against different numbers of initial clusters and the random seed (Section “Robustness” on page 7)
> > 2. The procedure is not very sensitive to the main hyperparameter in the objective: temperature (Appendix Fig. 14)
> > 3. Changing the digital twin architecture does not alter the results substantially (“Robustness” on p. 7 and Appendix Fig. 13)

---

### Official Review · Reviewer_KEew · 2023-10-31

**Soundness:** 3 good
**Presentation:** 3 good
**Contribution:** 3 good
**Rating:** 6
**Confidence:** 4

**Summary:**

This paper proposes maximize discriminative stimuli (MDS) that maximizely activates each functional cell types, and not activate the other types, and it is an extension of MEI (maximize excitation inputs). The algorithm follows the similar way as expectation maximization (EM), start from random assignment of groups, and maximize the activation objective in M step, and then reassign groups in E step. The total number of cell type groups is not defined as a prior, depends on the coverage and threshold of the EM algorithms. The work is evaluated on multiple real datasets and different animals (mouse retina, marmoset retina, and macaque V4). The proposed method is also helpful to provide time-efficient cell type identification.

**Strengths:**

Novelty and Importance:
1. The paper focus on a novel and important concept that finding inputs that maximize the difference of different functional cell types, which extend from single neuron MEI to group level.

Evaluations:
1. The work has been comprehensively evaluated on multiple experiments across multiple animals to demonstrate the soundness of the proposed approach. Sensitivity analysis including different initialization, number of clusters are included.

Writing:
1. The paper is well organized and presentation is good.

**Weaknesses:**

Method:
1. The clustering approach is based on the discreteness hypothesis of functional cell types, while ignoring the neurons might be close to the boundary or shows patterns that belongs to two functional cell types in different trials. Therefore, there might be limitation of separability.
2. It remains unclear how does the choice of neural networks or back-propagation algorithms affect the solutions.
3. Qualitatively, it is difficult to compare and interpret different MDS, and understand how they diverge from each other.

Baselines:
1. Simple baselines like searching nearest neighbor in original image space, or simple image interpolation could be introduced and compared.

Evaluation:
1. It seems that the optimized MDS has not been tested on real mouse yet, only evaluated on holdout neurons and trials.

**Questions:**

1. How does functional cell types here related to neuron classes of excitatory and inhibitory?
2. How to explain and interpret some off-diagonal entries in Fig 2D and Fig 4A?

---

> ### Author Response · Authors · 2023-11-12
>
> Thank you for your overall positive and constructive assessment of our paper.
>
> Before we address your concerns and questions, could you clarify your suggestion for simple baselines? We are happy to provide a baseline, but are unsure what you meant. How would you suggest to search for maximally discriminative stimuli by nearest neighbor search or image/video interpolation? Our goal is to find groups of neurons and corresponding stimuli that discriminate between these groups by maximizing/minimizing neural responses. What images/video would we use to start the nearest neighbor search and how would finding nearest neighbors of images lead to clusters of neurons?
>
> Regarding your concerns on the method:
>
> 1. You are right that discreteness is an underlying assumption. We consider this a feature rather than a problem: the motivation for developing our method was to be able to address the biological question whether neurons form discrete cell types that can be distinguished based on their functional properties. To answer this biological question, we need methods and models that capture certain underlying structural properties. Pitting such models against each other and testing how well they explain a dataset allows us to answer this question. We feel that ICLR as a machine learning venue is the right place for such method contributions and answering the biological question is beyond the scope of this venue.
>
> 2. This is a valid limitation that applies to all visualization techniques. The results indeed depend somewhat on architecture and optimizers. However, while the details differ, the features tend to be conceptually similar across hyperparameters (Walker et al., Nature Neuroscience 2019; Ustyuzhaninov et al., bioRxiv 2022, Figure 1F; Fu et al., bioRxiv 2023, Figure 2B left column). To what extent the differences are problematic depends on the underlying scientific question one tries to address. Therefore, while perfectly valid, we consider this concern to be somewhat orthogonal to the goal of our paper. We suggest adding a short discussion of this issue to the revised paper.
>
> 3. You are right that it might be difficult to compare and interpret the MDS. However, keep in mind that (a) in practice one would rarely use the MDS alone to characterize a neuron and (b) the MDS are still useful even if they are not interpretable at all, because experimentalists can use them to quickly determine a neuron’s type based on a small set of stimuli.
>
> Regarding evaluation: We agree that ultimately the method needs to be tested and validated in vivo. However, such experimental validation is challenging and beyond the scope of an ICLR paper where the focus is on the method development. We will add this great suggestion to the discussion of our paper.
>
> Regarding your questions:
>
> 1. Whether and how these clusters relate to excitatory vs. inhibitory cell types is an interesting question, but cannot be addressed with these datasets. The mouse retina dataset contains some displaced amacrine cells, but we restricted our analysis to the most widely available cell types in the dataset, which were all ganglion cells and, hence, excitatory. In the marmoset retina, we expect only ganglion cells. In the monkey V4 dataset we do not have information on cell types as they were recorded with extracellular multi-electrode recordings. We will revise the dataset description to mention this, accordingly.
>
> 2. The off-diagonal entries show that not all functional clusters can be separated perfectly by the method. This can be for two main reasons: (a) some cell types may not be identifiable by a single image/video (e.g. ON-OFF cells), which is a limitation of our method; (b) it is still an open question in neuroscience whether there are discrete cell types or whether there is a continuum of functional properties. Our method may contribute to answering this important and challenging question, but - as discussed above - actually answering the question is beyond the scope of an ICLR paper and ICLR is not the right venue for this.

---

> > ### Author Response · Authors · 2023-11-22
> >
> > Thank you once again for your encouraging feedback. We would like to follow up on your questions.
> >
> > You asked how the choice of neural networks affects our results: We now performed this experiment (Appendix A.6, Fig. 13) and showed that a modified digital twin architecture does not affect the results much. The resulting clusters and MDS look similar to those obtained with our initial architecture. Relatedly, we also addressed the robustness w.r.t. hyperparameters (Paragraph “Robustness” on p.7).
> >
> > Regarding your suggestion about a baseline in image space: We have now hypothesized about what you may have had in mind. One could replace the optimization of a maximally discriminative stimulus with a search for a maximally discriminative video snippet in our dataset for which we have measured neural responses. Unfortunately, such an approach would not be promising, because the video snippet from the mouse retina dataset (Höfling et al. bioRxiv 2022) most similar to a MDS had a cosine similarity of 0.4, for all other snippets it was below 0.2. This observation suggests that a search in image space for dataset examples is not very promising, because the dataset is too small. Using a larger dataset would not be useful either, because this would again require a predictive model and an exhaustive search through the dataset, which is likely to be more expensive than an optimization-based pre-image search as we perform it.
> >
> > You mentioned the importance of in-vivo evaluation. We now have added text to the Discussion, clearly stating that ultimately the method needs to be tested and validated in-vivo. In addition, we now mention that both the single cell MEIs for the digital twins and optimizer we used were successfully validated in-vivo (Höfling et al. bioRxiv 2022 and Willeke et al. bioRxiv 2023), suggesting our results might well generalize to an in-vivo experiment, too.
> >
> > Regarding your question on the interpretation of off-diagonal entries in Fig. 2: in our previous reply we mentioned two explanations; one of them is the limitation that some cell types may not be identifiable by a single image/video MDS (e.g. ON-OFF cells). We now extended our method to optimize two maximally discriminative stimuli instead of one, which revealed an additional cluster mainly associated with ON-OFF cells (Appendix A.5 and Fig. 12). This proof of principle shows that there is a path towards even more refined clustering based on our overall setup.

---

> > > ### Comment · Reviewer_KEew · 2023-11-22
> > >
> > > Thanks to the authors for their responses, and their efforts in adding new experiment results (another digital twin model, two MDS for explaining off-diagonal entries) and discussions (lack of validation from in-vivo experiments) to the revision.
> > >
> > > 1. The authors' new results on another digital twin model have partly addressed my concerns about **dependence on model choices**.  While I still encourage investigations on more diverse results from models trained with different architectures, objective functions, tasks, and learning rules in the future.
> > >
> > > 2. The authors correctly understand my proposed naive baseline - search instead of optimization for a maximally discriminative video snippet in the dataset, which might provide more interpretability as visualization in original image space rather than feature space. My concerns about this missing baseline have been addressed by the explanation of its limited performance, the small dataset size, and the time complexity of the exhaustive search. It would still be helpful if other stronger baselines were compared as Reviewer FrCb mentioned.
> > >
> > > 3. I am not fully convinced by the clarification of the discreteness hypothesis and adding biological insights is beyond the scope of ICLR. Is it possible to project MDS into a continuous space, and visualize the continuity and discreteness at the same time (i.e. whether there is a clear clustering pattern)? For example, boundary cells might raise an interesting point about the existence of **a continuum of functional properties** as the authors mentioned.
> > >
> > > 4. I agree with the authors' points that it could easily quickly determine a neuron’s type as the authors mentioned, while I think improving the lack of interpretability in MDS is still valuable.

---

> > > > ### Author Response · Authors · 2023-11-23
> > > >
> > > > Thank you very much for your encouraging response and for following up with such great observations. We will answer point by point:
> > > >
> > > > 1. We are pleased that our new experiments addressed your comment regarding particular digital twin choice. While previous studies successfully used a diverse set of digital twin architectures, objective functions, tasks and optimizers (e.g. Walker et al. 2019; Ustyuzhaninov et al., 2022; Fu et al., 2023, Tong et al., 2023) suggesting robustness across technical details, we agree that a separate paper, devoted to systematically studying this question, would be valuable future work.
> > > > 2. We are gratified that we could address your comment on the search-baseline. Based on Reviewer FrCb’s initial review, we clarified in our manuscript that we are indeed benchmarking our results against the empirically established reference cell types of Baden et al. (2016). After our clarifications, Reviewer FrCb agreed that this benchmark addressed their concerns and assessed our Baden et al. (2016) baseline comparison to be “valuable” and to “strengthen” our paper.
> > > > 3. You are right, MDS clustering is tailored to the common assumption of discrete cell types, akin to previous methods in the retina (Baden et al. 2016). Our results suggest a relatively clear clustering pattern (overall low off-diagonal elements in Figs. 2A, 4 and 5). We agree that the question about continuous vs. discrete functional properties is interesting and unsolved (Ustyuzhaninov et al. bioRxiv 2022), and that extending MDS to also capture continuous structures is an exciting direction for future work.
> > > > 4. We share the same thinking that interpretability is a valuable property. Our MDS visualize interpretable visual features that uniquely distinguish a cluster from all others. For example, in our marmoset results (Fig. 4), one can immediately understand only from the MDS which of the clusters is composed of ON, OFF, fast and slow cells. For more complex MDS as in area V4 (Fig. 5) we agree that interpretability is more difficult - this might be inherent to cells in this area that likely perform more complex computations. We agree that future work is needed to address this question, e.g. by improving the interpretability of feature visualizations in general.
> > > >
> > > > In summary, the observations you raised are important and interesting questions. While our revised manuscript addresses them to some extent, a full answer is not possible within the scope of our paper, and rather requires multiple future studies. We will explicitly state them in the final version of our paper.

---

### Official Review · Reviewer_FrCb · 2023-11-02

**Soundness:** 2 fair
**Presentation:** 2 fair
**Contribution:** 2 fair
**Rating:** 5
**Confidence:** 2

**Summary:**

Identifying and understanding cell types and their functional properties is crucial for understanding perception and cognition. Traditional methods have limitations, such as bias and lack of knowledge about functional cell types. A new approach, Maximally Discriminative Stimuli (MDS), uses optimization-based clustering with deep predictive models to identify functional cell types. This approach is successful across species and recording techniques and provides real-time cell type assignment, making experiments more efficient. MDS is interpretable and visualizes distinctive stimulus patterns for each neuron type.

**Strengths:**

* MDS provides a time-efficient on-the-fly cell type assignment by using a concise stimulus.
* MDS outperforms conventional approaches in identifying the correct cell type cluster, saving 20% of experimental time compared to traditional methods.

**Weaknesses:**

A potential weakness in the presented approach is that it assumes that the most informative stimuli for classifying cell types can be automatically chosen without requiring domain knowledge or expert input. While this is presented as an advantage, it may also be a limitation, as there could be cases where domain-specific insights are necessary for more accurate and nuanced cell type classification. Additionally, the success of the approach relies on the availability of a "digital twin" dataset, which may not always be readily available for all experimental studies, potentially limiting its applicability.

**Questions:**

While the authors express their belief in the usefulness of the algorithm and stimuli, there is no mention of empirical results or validation studies that demonstrate the actual effectiveness and applicability of the proposed approach.

One potential weakness of the clustering algorithm is its limitation in effectively distinguishing certain cell types with complex responses. For instance, for ON-OFF RGCs that respond to both light increments and decrements, optimizing a single, short MDS to maximize their response may not adequately capture their unique properties. Similarly, for cell types like "OFF suppressed 2," which exhibit a high baseline firing rate mostly suppressed by stimulation, the algorithm may not effectively identify them without specific adaptations, potentially requiring additional stimulus optimization strategies for better classification.

The absence of a benchmark is a significant weakness in this paper. It would be helpful if the author can add any performance to determine its effectiveness or limitations.

---

> ### Author Response · Authors · 2023-11-12
>
> Thank you for thoroughly reviewing our paper. We believe that some of your concerns might be based on a misunderstanding of the state of the field in neuroscience, but of course it is equally possible that we misunderstood your concerns. So please let us know if the explanations below address your concerns and consider updating your score.
>
>
> You point out the absence of empirical results (first question) and a benchmark (third question) as significant concerns, but we are unsure if we understand your suggestion correctly: For mouse retinal ganglion cells, Baden et al. (2016) established functional types based on a variety of functional and morphological properties. We tested our algorithm on a dataset labeled with those types and demonstrated good agreement of our clustering algorithm to the Baden et al. (2016) cell types (see first part of the Results and Figure 2). These results were empirically obtained on real retina responses and, in our opinion, represent exactly the benchmark you were asking for. Hence, we are unsure if we understood your main concern correctly. Could you clarify if this resolves your concern and what you suggest we do to increase the score you rated our paper with?
>
>
> Regarding your other questions and concerns:
>
>
> 1. Limitations: We agree that our algorithm is limited in distinguishing certain types with complex responses, e.g. ON-OFF cells from ON or OFF cells, which is why we discuss it explicitly in the discussion. Every method has limitations, including alternative methods that address similar problems (Ustyuzhaninov et al., bioRxiv 2022; Willeke et al., bioRxiv 2023). Note that ours is the first that has actually been validated on empirical data where something akin to biological ground truth exists (Baden et al. 2016). Beyond the retina (i.e. in visual cortex), we simply do not know whether functional types exist and, if so, how to identify them. Closing this gap is the key motivation behind our paper and, hence, an important contribution even if it has limitations. For sure, some of these limitations can be addressed, e.g. by extending our algorithm to generate multiple stimuli per cluster. Such improvements are interesting avenues for future work but beyond the scope of the current paper.
>
> 2. Domain specific knowledge: You are completely right - leveraging domain knowledge is of course helpful if it exists. In later processing stages such as V4, however, we simply do not know that much. In such situations we believe our approach is advantageous because it is still applicable. It is important to realize that one important use case of our method is to facilitate scientific discovery, i.e. specifically situations where the domain knowledge does not yet exist. We will improve the paper to more clearly discuss this.
>
> 3. Regarding the availability of digital twin models: Please note that the digital twin is needed only to find out what the cell types are. An important use case of our method is to quickly assign neurons to already identified types. In this situation, only the MDS are needed, but no digital twin. We will revise the paper to mention this explicitly.

---

> > ### Comment · Reviewer_FrCb · 2023-11-17
> > **RE:**
> >
> > Thank you for your detailed response to my review. I appreciate your efforts to address my concerns and provide clarifications. After carefully considering your explanations, I would like to discuss certain points further.
> >
> > Firstly, I appreciate the clarification on the empirical results and benchmarking. It is indeed valuable that you tested your algorithm on a dataset labeled with functional types from Baden et al. (2016), demonstrating good agreement in the clustering. This information was not explicitly highlighted in the initial paper, and I believe including these details in the Results section and Figure 2 strengthens your case. However, it might be beneficial to emphasize this point more prominently in the manuscript to ensure readers easily grasp the empirical validation aspect.
> >
> > Concerning the limitations, I understand that every method has its constraints, and it's positive that you openly acknowledge them in the discussion. The comparison to alternative methods, such as Ustyuzhaninov et al. (bioRxiv 2022) and Willeke et al. (bioRxiv 2023), adds context to your work. I suggest reinforcing this comparison and explicitly stating how your method stands out in terms of empirical validation and applicability in situations where domain knowledge is limited. Additionally, your proposed future improvements, such as extending the algorithm to generate multiple stimuli per cluster, could be briefly outlined as avenues for addressing some of the identified limitations.
> >
> > Regarding domain-specific knowledge, I appreciate your perspective on the applicability of your approach in situations where limited knowledge exists. Emphasizing this point in the manuscript, perhaps in the Introduction or Methodology sections, will help readers understand the unique contribution your method makes in scenarios with sparse domain knowledge.
> >
> > Finally, your clarification on the digital twin models is helpful. Clearly stating in the paper that the digital twin is only needed for identifying cell types, while the MDS alone suffices for assigning neurons to identified types, will improve the paper's clarity.
> >
> > In summary, I suggest incorporating these additional details and emphases into the manuscript to enhance its overall clarity and strengthen the case for your method. I appreciate your responsiveness to my feedback and look forward to the revised version of your paper.

---

> > > ### Author Response · Authors · 2023-11-22
> > >
> > > Thank you very much for your encouraging response. We appreciate your feedback and have now incorporated all of your points into our revised manuscript, substantially improving clarity. Specifically, we have made the following changes:
> > >
> > > - Added text to explicitly highlight the benchmarking of our clustering algorithm on empirical data in the Introduction, first part of the Results, caption of results Fig. 2, and first part of the Discussion.
> > > - Added paragraphs to the Discussion (at “Data-driven, interpretable cell type discovery with MDS.”) explicitly comparing our clustering method to Ustyuzhaninov et al. (bioRxiv 2022) and Willeke et al. (bioRxiv 2023). We now highlight that our clustering algorithm is the first that has actually been validated on empirical data.
> > > - Added text outlining the extension of our algorithm to multiple stimuli to the Discussion and a pilot experiment to the Appendix (see further details below).
> > > - Added emphasis on the applicability of our method in cases where no domain knowledge exists to the Introduction and Discussion.
> > > - Made it more explicit in Abstract, Results and Discussion that the digital twin is needed only once for identifying the cell types, while afterwards, the MDS alone suffices for assigning neurons to identified types.
> > >
> > > In your initial review and your response, you were interested in how our method could be extended to distinguish ON-OFF cells from ON or OFF cells by optimizing multiple MDS at the same time - an exciting idea we now would like to follow up on. We extended our method to optimize two maximally discriminative stimuli instead of one, which revealed an additional cluster mainly associated with ON-OFF cells (Appendix A.5 and Fig. 12). This proof of principle shows that there is a path towards even more refined clustering based on our overall setup.
> > >
> > > Thank you once again for mentioning these points, which now have significantly strengthened the clarity and exposition of our results.

---

### Official Review · Reviewer_69wH · 2023-11-06

**Soundness:** 3 good
**Presentation:** 3 good
**Contribution:** 2 fair
**Rating:** 6
**Confidence:** 2

**Summary:**

In this paper, the authors developed a framework to jointly cluster cells into functional cell types and obtain Maximally Discriminative Stimuli (MDS) for each functional cluster, based on an EM-type iteration. The MDS aims to stimulate one cell type while suppressing all others, as an extension to previously developed Maximally Exciting Inputs (MEIs). The authors carried out several real data analysis to demonstrate the capability of the proposed framework.

**Strengths:**

The paper is relatively easy to follow and well-structured. The method seems to be straightforward and intuitive. The authors provided multiple experiments to demonstrate the performance of the proposed framework.

**Weaknesses:**

My primary concern is how useful and interpretable the method will be for actual practice. The experiments essentially treated a carefully-examined existing publication as the ground truth to compare with. I assume in a lot of real scenarios, we might already have this kind of biological baselines. For cases where they are not available, the interpretability of the identified MDS might be important.

**Questions:**

1. For the sub-cluster split, the authors used an "evaluation MDS" for determining if clusters should be kept. This seems to be a newly initialized, separately trained MDS which does not necessarily tell whether the sub-clusters are bringing in better cell type identification. I hope the authors can explain more about the logic of this.
2. Can the authors elaborate more on Figure 2B? They referenced figure 2B when stating in the text that “none of the MDS exhibited direction selectivity”, but I didn’t quite understand why Figure 2B is referenced here.
3. For the experiments in Figure 2, the identified MDS seems to be largely combination of several functional cell types from the published Baden et al (2016) paper. I wonder if further tweaking with the sub-clustering procedure can better recover the published functional cell types.
4. How will the result change if the number of gradient ascent steps is changed during each M-step? Trying to understand if there is a delicate balance needed between how well the M-step is optimized and the E-step, or if the procedure is relatively robust and almost guaranteed to converge to the same set of results.

---

> ### Author Response · Authors · 2023-11-12
>
> Thank you for your encouraging feedback on our paper. We would like to start by addressing your main concerns about usefulness in practice and interpretability of the method. Please let us know if this addresses your concern or what you believe we should do or demonstrate to increase your score.
>
> Regarding usefulness: As you state correctly, we used the cell types by Baden et al. (2016) to show our algorithm’s effectiveness. However, your assumption that the same biological baselines exist in a lot of real scenarios is not true. Beyond the retina (i.e. in visual cortex), it is not known if cells can be clustered into functional types and, we do not know how many types exist. Additionally, it is not clear how exactly to identify them because the method used by Baden et al. (2016) does not generalize to visual cortex. Closing this gap is the key motivation behind our paper: Our clustering approach finds optimal stimuli to determine functional types and assigns neurons quickly to their type.
>
> Regarding interpretability: Currently there exist no experimentally validated methods. The methods that do exist use either in-silico experiments with simple, classical stimuli (Ustyuzhaninov et al., bioRxiv 2022; but this approach does not generalize to lesser known areas such as V4) or similarity of single cell maximally exciting images (Willeke et al, bioRxiv 2023). In contrast to these approaches, our method offers the added value that the stimuli we obtain are not only highly activating, but actually visually what feature a cell type “specializes” on that other cell types do not respond to. Thus, our method offers potential to drive the field forward exploring functional cell types for many real world scenarios where no clustering baselines exist (e.g. Ustyuzhaninov et al., bioRxiv 2022; Willeke et al, bioRxiv 2023).
>
> We will revise the paper to discuss these points more clearly.
>
> Regarding your questions:
>
> 1. After running EM clustering on the sub-cluster split, we evaluated if splitting improves overall clustering. Therefore, we kept neuron’s assignments to the main clusters and new sub-cluster candidates, but randomly initialized the stimulus associated with each of them. Next, we optimized the MDS for all these clusters and re-assigned neurons based on the optimized MDS. Now, neuron assignments and MDS reflected the overall clustering state - including the subclusters - which allowed us to precisely estimate if splitting improved overall clustering. We will clarify this in the paper.
>
> 2. We agree, in the first principle component (Figure 2B) direction selectivity cannot be examined. The digital twin model (Höfling et al. bioRxiv 2022) used space-time separated convolution learning a time-trace and spatial filter independently (similar as depicted in Figure 2B), and hence was not able to learn direction selectivity. We apologize for referencing Figure 2B for such claims and will revise the paper accordingly.
>
> 3. Baden et al. (2016) relied on functional responses to chirp and moving bar stimuli and morphological properties. As the digital twin cannot model direction selectivity or morphological properties, we removed these features from the classifier that was used to label our dataset, showing degraded separability across the types reported in Baden et al. (2016) (Figure 2E). Hence, the digital twin lacking these features poses a limit on the granularity of our clusters.
>
> 4. In the M-step we optimize MDS until the objective does not change significantly anymore to yield stable evaluations of our objective (Eq. 1). During algorithm development for the mouse retina data we verified sufficient training steps were performed by inspecting the objective values across steps. For marmoset and monkey V4 data, we automatically checked for convergence: we stopped MDS optimization if an exponential moving average of the objective did not increase anymore for 100 optimization steps and verified from the objective values across optimization steps that this stopping criterion led to a reasonably converged objective. We now have realized this information was not clearly mentioned throughout the paper, which we will revise accordingly.

---

> > ### Author Response · Authors · 2023-11-22
> >
> > We added another experiment that pertains to your third question whether further tweaking could better recover the functional cell types published by Baden et al. (2016): We extended our method to optimize two maximally discriminative stimuli instead of one, which revealed an additional cluster mainly associated with ON-OFF cells (Appendix A.5 and Fig. 12). This proof of principle shows that there is a path towards even more refined clustering based on our overall setup.
> >
> > Your initial review prompted us to clarify the issues of usefulness and interpretability of our method. We discussed these in our earlier reply to your review above. As promised, we also revised Introduction and Discussion in our manuscript accordingly to include those important points. Specifically, the main text now states more explicitly why MDS clustering is more interpretable than existing approaches, and that there are common situations (e.g. little domain knowledge) where MDS clustering enables data-driven, unbiased discovery of functional cell types. Again, thank you for raising these points - they helped us to improve the clarity of our manuscript.

---

### Author Response · Authors · 2023-11-22
**Revised version with new experiments and improved text**

Thanks to all reviewers for their helpful comments. We have incorporated all suggestions, which has strengthened the paper substantially. Here are the main changes:

- New results (Section “Robustness”, p.7) on the robustness of the approach w.r.t. changes in hyperparameters:
   1. temperature (Fig. 14)
   1. number of initial clusters (Figs. 6, 9, 11)
   1. multiple runs (Fig. 7)
   1. digital twin architecture (Fig. 13).
- New pilot experiment showcasing that our algorithm can be extended to optimize two MDS at the same time, which improves in discriminating ON-OFF cells from ON and OFF cells (Appendix A.5 and Fig. 12).
- Revised many parts of the manuscript to clarify descriptions, addressing all reviewer comments.
- Streamlined the text throughout the paper, improving the clarity and presentation of our results.
- During revision, we found a small bug in how we aligned the receptive field positions for the marmoset digital twin. We have now fixed it and found even stronger clustering results with fewer and more discernible clusters (Fig. 4).

---

> ### Author Response · Authors · 2023-11-23
> **Revised version with new reviewer suggestions and improved text**
>
> Once more, we want to thank all reviewers for their helpful comments and replies. In another revision of the manuscript that we just uploaded, we have incorporated new suggestions of reviewer KEew, smoothed text, and improved the structure of the Discussion section.

---

### Meta-Review · Area_Chair_rNrj · 2023-12-12

**Metareview:**

This paper describes a novel method for identifying functional cell types in the retina and visual cortex using a novel paradigm that involves alternating between clustering and synthesizing "Maximally Discriminative Stimuli" to improve information about the clusters.  This paper generated considerable interest and discussion among the reviewers, and was among the more original and interesting papers I examined this year.  The reviewers raised concerns about interpretability and the usefulness of the algorithm in real-world settings, and were split about whether it should be accepted to the meeting in its current form. Ultimately, I felt that reviewer concerns about an assumption of discrete cell classes, or about needing to incorporate domain information, were not so severe as to outweigh the paper's strengths.  (In fact, I would regard the fact that the algorithm does not need to domain information as a positive). Thus, I feel the paper should be accepted.  I congratulate the authors on the thoroughness of their replies to reviewer concerns, and would ask them to revise the final paper to address all reviewer comments clearly.

**Justification For Why Not Higher Score:**

Scores were very borderline, so if anything it should be moved down, not up.

**Justification For Why Not Lower Score:**

I felt the several concerns raised by reviewers (eg, that the method presupposes the existence of discrete cell classes, or does not require domain information) to be somewhat lacking in relevance.  The paper seems highly original and likely to elicit a high level of interest at the meeting (and in future literature), thus I am happy to see it accepted even if the experimental validation is still somewhat underpowered.

---

### Decision · Program_Chairs · 2024-01-16

Accept (poster)